# Cold-activated brown fat-derived extracellular vesicle-miR-378a-3p stimulates hepatic gluconeogenesis in male mice

Jinhong Xu[1,7], Le Cui[1,7], Jiaqi Wang[1,7], Shasha Zheng[1,7], Huahua Zhang[1], Shuo Ke[1], Xiaoqin Cao[1], Yanteng Shi[1], Jing Li[1], Ke Zen [1], Antonio Vidal-Puig [2,3] ✉, Chen-Yu Zhang [1,4,5,6] ✉, Liang Li [1,4] ✉ & Xiaohong Jiang [1,3,4] ✉

During cold exposure, activated brown adipose tissue (BAT) takes up a large amount of circulating glucose to fuel non-shivering thermogenesis and defend against hypothermia. However, little is known about the endocrine function of BAT controlling glucose homoeostasis under this thermoregulatory challenge. Here, we show that in male mice, activated BAT-derived extracellular vesicles (BDEVs) reprogram systemic glucose metabolism by promoting hepatic gluconeogenesis during cold stress. Cold exposure facilitates the selective packaging of miR-378a-3p—one of the BAT-enriched miRNAs—into EVs and delivery into the liver. BAT-derived miR-378a-3p enhances gluconeogenesis by targeting p110α. miR-378 KO mice display reduced hepatic gluconeogenesis during cold exposure, while restoration of miR-378a-3p in iBAT induces the expression of gluconeogenic genes in the liver. These findings provide a mechanistic understanding of BDEV-miRNA as stress-induced batokine to coordinate systemic glucose homoeostasis. This miR-378a-3p-mediated interorgan communication highlights a novel endocrine function of BAT in preventing hypoglycemia during cold stress.

Brown adipose tissue (BAT) is the primary site of non-shivering thermogenesis (NST), which protects mammals from hypothermia[1]. NST is significantly increased in response to cold exposure and is mediated by the increased mitochondria uncoupling within BAT dissipating energy to produce heat[2,3]. To replenish intracellular energy stores, cold-activated BAT requires sharply increased uptake of glucose and fatty acids as fuel supplies to sustain high levels of mitochondrial respiration. In rodents, activated BAT is estimated to clear up to 75% and 50% of glucose and triglycerides from circulation[4]. As landmark papers have demonstrated that active BAT is present in adult humans when exposed to cold, studies have been specifically designed to investigate the role of BAT in glucose metabolism[5,6]. [18]F-fluorodeoxyglucose PET ([18]F-FDG-PET) imaging showed that glucose uptake was increased 12-fold in cold-induced BAT in humans[7]. Given this significant role of activated BAT in consuming glucose, a recent study showed that BAT mediates tumour suppression by competitive use of blood glucose[6]. Notably, cold exposure did not affect whole-body glucose metabolism in men who did not have detectable BAT, indicating a physiologically significant role of activated BAT in regulating whole-body glucose homoeostasis[7].

[1]Nanjing Drum Tower Hospital Center of Molecular Diagnostic and Therapy, State Key Laboratory of Pharmaceutical Biotechnology, School of Life Sciences, Nanjing University, Nanjing, Jiangsu, China. [2]Wellcome-MRC Institute of Metabolic Science, Addenbrooke's Hospital, University of Cambridge Metabolic Research Laboratories, Cambridge, UK. [3]Cambridge University Nanjing Centre of Technology and Innovation, Nanjing, China. [4]Jiangsu Engineering Research Center for MicroRNA Biology and Biotechnology, NJU Advanced Institute of Life Sciences (NAILS), Nanjing, Jiangsu, China. [5]Research Unit of Extracellular RNA, Chinese Academy of Medical Sciences, Nanjing, Jiangsu, China. [6]Institute of Artificial Intelligence Biomedicine, Nanjing University, Nanjing, Jiangsu, China. [7]These authors contributed equally: Jinhong Xu, Le Cui, Jiaqi Wang, Shasha Zheng. ✉e-mail: ajv22@medschl.cam.ac.uk; cyzhang@nju.edu.cn; liangli@nju.edu.cn; xiaohongjiang@nju.edu.cn

Mammals can maintain a narrow and constant blood glucose range under different physiological conditions, which requires an intricate balance between glucose uptake and endogenous glucose production[8]. The liver plays a significant role in maintaining glucose homoeostasis because it stores glucose postprandially and produces it during fasting[9]. Hepatic glucose production (HGP) is crucial for systemic glucose homoeostasis and accounts for ~90% of endogenous glucose production[10,11]. HGP is markedly upregulated in the fasted state due to increased hepatic glycogenolysis and gluconeogenesis, thereby providing glucose as a fuel supply for other glucose-consuming tissues, including the brain, kidney, and blood cells[12,13]. Cold and fasting are the two fundamental challenges in mammals during evolution; however, most studies have focused on the importance of gluconeogenesis during fasting. Few studies have reported that cold stress also promotes gluconeogenesis and causes profound metabolic changes in the liver, but the role and physiological impact of the liver in cold adaptation remains unclear[14–16].

BAT has received much interest since the discovery of active brown fat in adults, which takes up glucose avidly upon cold stimulation in humans[7,17]. In addition to its role in thermogenesis, evidence suggests that BAT also functions as an endocrine organ[18]. BAT secretes peptidic and nonpeptidic molecules, including bioactive lipids and miRNAs, also called batokines. Like white adipokines, batokines also act on other metabolic tissues, including the brain, pancreas, liver, and bone, thereby contributing to energy homoeostasis[19]. Wang et al. showed that BAT-released NRG4 directly targeted the liver and attenuated hepatic lipogenic signalling to protect against hepatic steatosis[20]. More recently, Sugimoto et al. showed that BAT-derived MaR2 contributes to the cold-induced resolution of inflammation[21]. These studies underscore the profound influences of batokines on liver metabolism and homoeostasis.

MicroRNAs, posttranscriptional modulators of gene expression, are critical regulators of diverse physiological and pathological processes[22,23]. In addition to residing in cells, miRNAs are detected in various body fluids as circulating miRNAs[24,25]. Growing evidence indicates that most miRNAs stably expressed in circulation are either part of a complex with Ago2 or within extracellular vesicles (EVs)[26,27]. EVs, double membrane-bound vesicles containing RNAs, proteins and lipids, are indispensable endocrine mediators between different cells[28–30]. By transplanting BAT into a Dicer KO mice, Thomou et al. reported that BAT-derived EV-contained miR-99b reduced hepatic *Fgf21* mRNA and improved glucose tolerance[31]. Notably, Chen et al. reported that BAT's activation significantly changed the EV miRNAs' pattern and that miR-92 was inversely correlated with BAT activity, suggesting that stress signals dynamically regulated BAT-derived EV miRNAs[32]. However, the significance of EV-containing miRNA-mediated metabolic signalling in cold-induced BAT physiology is poorly understood. In light of its ability to actively use glucose and lipids for heat generation, we hypothesised that the EV-miRNAs secreted by cold-activated BAT might support fuel availability and coordinate BAT activity with systemic carbohydrate metabolism.

Here, we report that in male mice, cold-activated BAT-derived extracellular vesicles (BDEVs) reprogram systemic glucose metabolism by promoting hepatic gluconeogenesis during cold stress. Cold exposure stimulates the selective packaging of miR-378a-3p—one of the BAT-enriched miRNAs—into EVs and delivery into the liver, promoting gluconeogenesis by targeting p110α. Thus, we demonstrate that cold-induced BDEV-containing miR-378a-3p could serve as an endocrine signalling molecule to regulate hepatic gluconeogenesis under cold stress.

## Results

### Cold-activated BAT facilitates hepatic gluconeogenesis

Age-matched C57BL/6 J male mice were subjected to cold (4 °C) for 0, 24, 48 and 72 hours (h). Cold exposure promoted an early fall in body temperature (4 h) in mice, but the mice rapidly adapted, and body temperature returned to the control level, indicating activation of thermogenesis (Fig. 1a). Considering that BAT is one of the most important organs for generating heat in mice, we assessed the expression of uncoupling protein 1 (UCP1), a hallmark of BAT activation. The increased BAT activity was evident by the denser appearance of interscapular BAT (iBAT) in mice, along with significantly increased UCP1 protein level after 72 h of cold exposure (Fig. 1b, c and Supplemental Figure S1a). Consistently, the mRNA levels of thermogenic-associated genes, including *Pgc-1α*, *Dio2*, and *Elovl6*, were profoundly increased in cold-induced BAT (Fig. 1d). In line with previous studies[21], cold exposure markedly stimulated daily food intake while decreasing body weight in mice (Fig. 1e, f), suggesting a boost of metabolic rate to defend against hypothermia. As evidenced by the significantly elevated 2-NBDG uptake and increased glucose transporter GLUT4 translocation to plasma membranes in cold-activated BAT (Fig. 1g, h), cold exposure robustly increased glucose uptake in BAT. Interestingly, on the other hand, the hepatic gluconeogenesis was also significantly increased in the cold exposed mice, as indicated by the markedly induced glucogenic genes in the liver, including *Pgc-1α*, *Pepck* and *G6Pase* (Fig. 1i). In addition, the phosphorylation of AKT, FOXO1 and GSK3β was markedly suppressed by cold exposure in liver, suggesting that cold exposure showed coordinated effects on BAT (elevated glucose uptake) and liver (increased gluconeogenesis) glucose metabolism (Fig. 1j and Supplemental Figure S1b). As gluconeogenesis is typically engaged in hypoglycaemia conditions, such as fasting, we hypothesized that the BAT activation was coordinating stress physiology to fuel the increased energy demand during cold exposure. To test whether the effect of cold exposure on hepatic glucose metabolism could be directly influenced by BAT, the interscapular BAT (iBAT) depot was removed from mice (BATectomy) (Fig. 1k). Compared with the sham-operated group, a sharper early fall of body temperature was observed in BATectomy mice (4 h), along with a relatively slower adaptation (Fig. 1l). Compared with sham-operated controls, BATectomy did not alter food intake, but resulted in increased blood glucose level (Fig. 1m, n). Meanwhile, decreased gluconeogenic gene expression levels (Fig. 1o), along with increased phosphorylation of AKT, FOXO1 and GSK3β was observed in the liver upon cold exposure (Fig. 1p and Supplemental Figure S1c). Taken together, these data strongly suggest that BAT activation may directly influence liver glucose metabolism by facilitating hepatic gluconeogenesis during cold stress.

### Cold stress promotes BAT-derived extracellular vesicle secretion and delivery into the liver

In addition to NST, emerging evidence suggests that BAT also functions as an endocrine organ. Here, we asked if the BAT-secreted factors, such as extracellular vesicles (EVs), might also play a role in this BAT-liver crosstalk in addition to classical hormonal regulation. Since chronic cold exposure (longer than 7 days) may result in the expansion of BAT in mice, the iBAT weight was assessed. Although 72 h of cold exposure robustly activated the thermogenic activity of iBAT (Fig. 1b), its weight was not markedly changed yet (~100 mg, Supplementary Figure S2a). As indicated by transmission electron microscopy (TEM) images, BAT-derived extracellular vesicles (BDEVs) were isolated from the iBAT (100 mg, approximately the amount from each mouse) culture medium via ultracentrifugation from mice housed at room temperature (RT) or cold-exposed for 72 h (Fig. 2a, b). As revealed by nanoparticle tracking analysis (NTA), the collected BDEVs amount gradually increased within 24 h and reached a plateau in both the control (RT-BAT) and cold exposed (Cold-BAT) group, thus, a 24 h incubation period was chosen for further studies (Supplementary Figure S2b). Of note, the Cold-BAT generally displayed more EV secretion than RT-BAT ($4 \times 10^9$ RT-BDEV particles, ~80 μg RT-BDEV protein; while $1 \times 10^{10}$ Cold-BDEV particles, ~160 μg Cold-BDEV protein

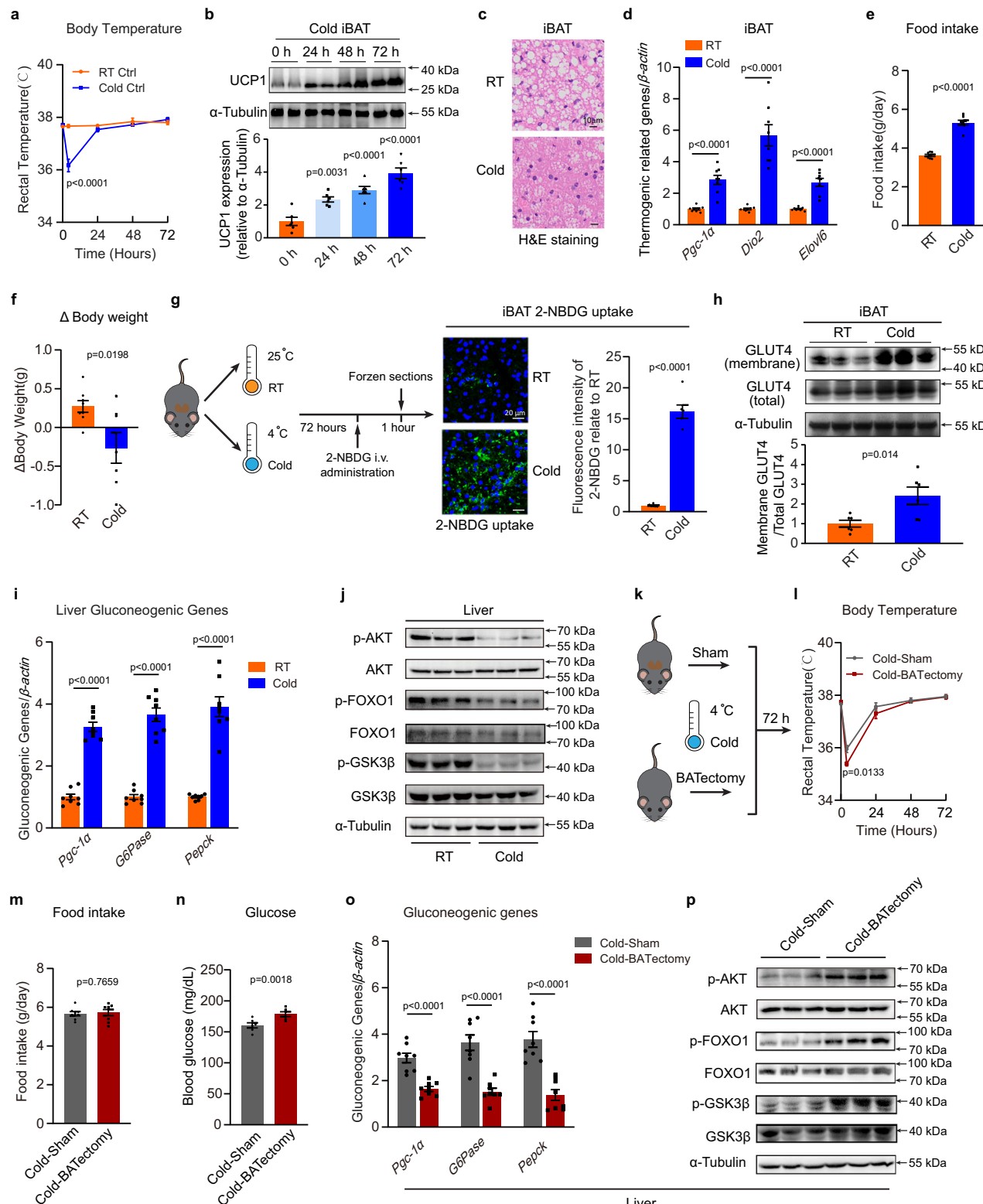

collected at 24 h, respectively) (Fig. 2c, d and Supplementary Figure S2c). Western blot analysis showed that LAMIN A/C, which found abundantly in the nucleus, was barely detected in the RT-BDEVs, consistent with the known characteristics of EVs[33]. UCP1, which is enriched in the mitochondria inner membrane protein of BAT, was detected in medium/lagre EV fractions. However, its detection in the same quantify of RT-BDEV samples was limited (Supplementary Figure S2e). More importantly, the protein levels of EV markers, including CD9, CD63 and ALIX were markedly increased in cold-activated BDEVs compared with EVs isolated from the same amount of control BAT (Fig. 2e and Supplemental Figure S2d). These data strongly suggest that cold exposure activates BAT thermogenesis and promotes BDEV secretion. PKM2, an isoenzyme of the glycolytic enzyme pyruvate kinase, was reported to promote tumour cell EV secretion[34]. We found that the phosphorylation of PKM2 (p-PKM2) and total PKM2 protein levels were significantly increased in cold-activated BAT, indicating

**Fig. 1 | BAT stimulates hepatic gluconeogenesis upon cold exposure. a–j** Eight-week-old male C57BL6/J mice were placed in RT (25 °C) or cold (4 °C) for 72 hours and fasted for 4 hours before sacrifice. **a** Rectal temperature at 0, 4, 24, 48, and 72 h (*n* = 6 biological replicates, from 2 independent experiments). **b** Western blotting and densitometry analysis of UCP1 protein levels in iBAT (*n* = 6 biological replicates, from 3 independent experiments). **c** H&E staining, scale bars:10 μm. (*n* = 6 each group, 3 random fields per sample, representative images were shown). **d** Relative expression of thermogenic-related genes in iBAT, **e** food intake and **f** body weight change (*n* = 8 biological replicates, from 2 independent experiments). **g** Experimental schematic (left panel), fluorescence image of 2-NBDG uptake into iBAT (medium panel, scale bars: 20 μm), and densitometry analysis of positive signals (right panel) (*n* = 3 each group, 5 random fields per sample, representative images were shown). **h** Western blotting and densitometry analysis of GLUT4 expression in the membrane and total fractions of iBAT. (*n* = 6 each group, from 2 independent experiments). **i** Relative expression of gluconeogenic-related genes in

the liver (*n* = 8 each group, from 2 independent experiments). **j** Western blotting analysis of total and phosphorylated AKT, FOXO1 and GSK3β in liver. (*n* = 6 for each group, from 2 independent experiments). **k–p** Eight-week-old male C57BL6/J mice underwent BATectomy or a sham procedure. Following a 10-day recovery period, the mice were placed in cold for 72 h and fasted for 4 h before sacrifice. **k** Experimental schematic. **l** Rectal temperature at 0, 4, 24, 48, and 72 h (*n* = 6 biological replicates, from 2 independent experiments). **m** Food intake, **n** blood glucose and **o** relative expression of gluconeogenic-related genes in liver (*n* = 8 each group, from 2 independent experiments). **p** Western blotting analysis of total and phosphorylated AKT, FOXO1 and GSK3β in liver. (*n* = 6 each group, from 2 independent experiments). Data presented as mean ± sem. Statistical analysis was performed two-way ANOVA followed by Bonferroni's multiple comparisons test **a**, **l**, one-way ANOVA followed by Bonferroni's multiple comparisons test **b** and others performed two-sided unpaired t-test. Source data are provided as a Source Data file.

that PKM2 may also play a critical role in promoting EV release from activated BAT (Fig. 2f). Next, for in vitro BDEV tracking, $3.5 \times 10^7$ BDEVs were labelled with the fluorescent dye Dil and incubated with $1.5 \times 10^5$ hepatocytes. As indicated by the red fluorescence staining of the hepatocytes, fluorescent signals could be detected in hepatocytes as early as 2 h post incubation, and gradually increased and reached a plateau phase after 6 h of incubation, which exhibited efficient uptake of BDEVs (Fig. 2g and Supplemental Figure S2f). For direct in vivo visualization of EV-mediated communication between BAT and the liver under cold stress, we constructed reporters to label EVs to track activated BAT-derived EV release and uptake. Briefly, enhanced green fluorescence protein (EGFP) was fused with a palmitoylation signal (Palm-EGFP), and the fusion protein was constructed in a plasmid for double-membrane labelling, including cell and EV membranes (Fig. 2h). The primary cultured brown adipocytes were transduced with a lentivirus vector encoding either GFP (control) or Palm-GFP to examine EV reporter expression. Confocal microscopy of brown adipocytes showed that Palm-GFP preferentially labels the membrane structure of the cells, including the cell membranes, whereas normal Ctrl-GFP was only expressed within the cytoplasm of the control cells. Notably, unlike the Ctrl-BDEVs, green fluorescence was also observed in EVs isolated from Palm-GFP⁺ transduced cells (Palm-BDEVs), confirming that Palm-GFP labelling can be used to track EVs (Fig. 2h). Then, the Palm-GFP was constructed under the adipose tissue-specific promoter FABP4 and introduced into the iBAT of mice. As the schematic diagram shows in Fig. 2i, by using a multipoint injection, $10^8$ lentiviral transduction vectors (TUs) were inoculated twice into BAT at the indicated time points, and the mice were cold-exposed prior to sacrifice. In accordance with previous results (Fig. 1b and 1c), coimmunostaining of UCP1 (BAT marker, red) and GFP (green) confirmed that cold exposure resulted in a denser appearance and higher UCP1 expression of iBAT, with approximately 70% BAT infection in both groups (Fig. 2j). Notably, the appearance of GFP signals in the liver suggested that cold exposure resulted in significantly increased BDEV uptake by the hepatocytes (CK18-positive, red) compared to RT (Fig. 2k and Supplemental Figure S3a). The GFP mRNA was not detected in the liver of mice infected with Palm-GFP lentivirus (Supplementary Figure S3b). Moreover, neither the fluorescent signal nor mRNA expression was detected in the liver infected with Ctrl-EGFP, indicating that it is unlikely that the GFP signal detected in the liver was due to viral leakage (Supplementary Figure S2g-S2i). Collectively, through visualization and tracking of BDEVs, we showed that cold exposure facilitates BDEV secretion and delivery into the liver. In addition, we also checked if Cold-BDEVs also reach other peripheral metabolic tissues and the brain. Although the fluorescent GFP signals were highly expressed in the liver, the signals could also be detected in the kidney, heart, white adipose tissue and skeletal muscle but not the brain (arcuate nucleus of the hypothalamus, hippocampus and cortex) (Supplementary Figure S3c).

## Cold-activated BDEVs enhance gluconeogenesis

Next, we asked whether the EVs secreted by cold activated BAT could serve as endocrine modulators to regulate liver glucose metabolism. Since cold exposure significantly increased BDEVs secretion, for 24 h incubation, $4 \times 10^9$ RT-BDEV particles (~ 80 μg RT-BDEV protein) and $1 \times 10^{10}$ Cold-BDEV particles (~ 160 μg Cold-BDEV protein) were isolated from the medium from 100 mg RT-BAT and Cold-BAT after 24 h of incubation, respectively (Fig. 2d, Supplementary Figure S2c). Thus, to mimic physiological conditions, $1.6 \times 10^9$ RT-BDEVs (~32 μg) and $4 \times 10^9$ Cold-BDEVs (~64 μg) (~1/5 of the total EVs isolated from cultured RT-BDEVs or Cold-BDEVs × 2 days) were administered to $4 \times 10^6$ primary cultured hepatocytes (~1/5 hepatocytes isolated from each mouse liver), respectively. The BDEVs and hepatocytes were co-incubated for 48 h, followed by functional analysis (Supplementary Figure S4a). Compared with the cells treated with $1.6 \times 10^9$ RT-BDEVs (~32 μg), the phosphorylation level of AKT was dramatically decreased in the hepatocytes administered $4 \times 10^9$ Cold-BDEVs (~64 μg), concurrent with markedly increased gluconeogenic genes (*Pgc-1α, G6Pase, Pepck*) (Supplementary Figure S4b-S4c). To explore whether treatment with equal amounts of Cold-BDEV particles or protein would also have similar effects, hepatocytes were incubated with $1.6 \times 10^9$ Cold-BDEVs (equal particle numbers as RT-BDEVs, Supplementary Figure S4a-S4c) or 32 μg Cold-BDEVs (equal amounts RT-BDEV proteins, Fig. 3a–d), respectively. Both treatments also significantly increased glucose production, along with increased gluconeogenic genes (*Pgc-1α, G6Pase, Pepck*) and reduced phosphorylation levels of AKT. Given the marked in vitro effects of cold-activated BDEVs, in vivo studies were conducted for further validation. As shown in Fig. 3e, mice housed at RT were injected intravenously with Cold-BDEVs or RT-BDEVs. Equal protein amounts of RT-BDEVs and Cold-BDEVs (80 μg each day) were used first. Compared with the RT-BDEV administered group, Cold-BDEV treatment led to increased levels of gluconeogenic gene expression and decreased levels of phosphorylated AKT, FOXO1 and GSK3β (Fig. 3h, i). Notably, Cold-BDEV administration did not alter food intake or insulin or glucagon levels in mice (Fig. 3j–l). This finding was supported by pyruvate tolerance tests (PTT) in Cold-BDEV treated mice, where pyruvate conversion to glucose was increased compared with mice administered with RT-BDEV (Fig. 3f, g). Similar effects were observed in mice treated with equal RT- and Cold-BDEV particles (~4 × 10⁹ particles each day). Furthermore, for those mice treated with BDEVs secreted from the same amount (100 mg) of cultured RT-BAT (~4 × 10⁹ RT-BDEVs) and Cold-BAT (~ 1 × 10¹⁰ Cold-BDEVs), a more significantly enhancement of hepatic gluconeogenesis was observed in mice treated with Cold-BDEVs (Supplementary Figure S4d-S4f). To verify the critical role of Cold-BDEVs in regulating gluconeogenesis, the EV secretion inhibitor GW4869 was used to block cold-induced BDEV production. As indicated in Fig. 3m, the cold-exposed mice were injected in situ with either GW4869 or a control vehicle three times. BAT local GW4869 treatment did not affect mouse food intake, insulin

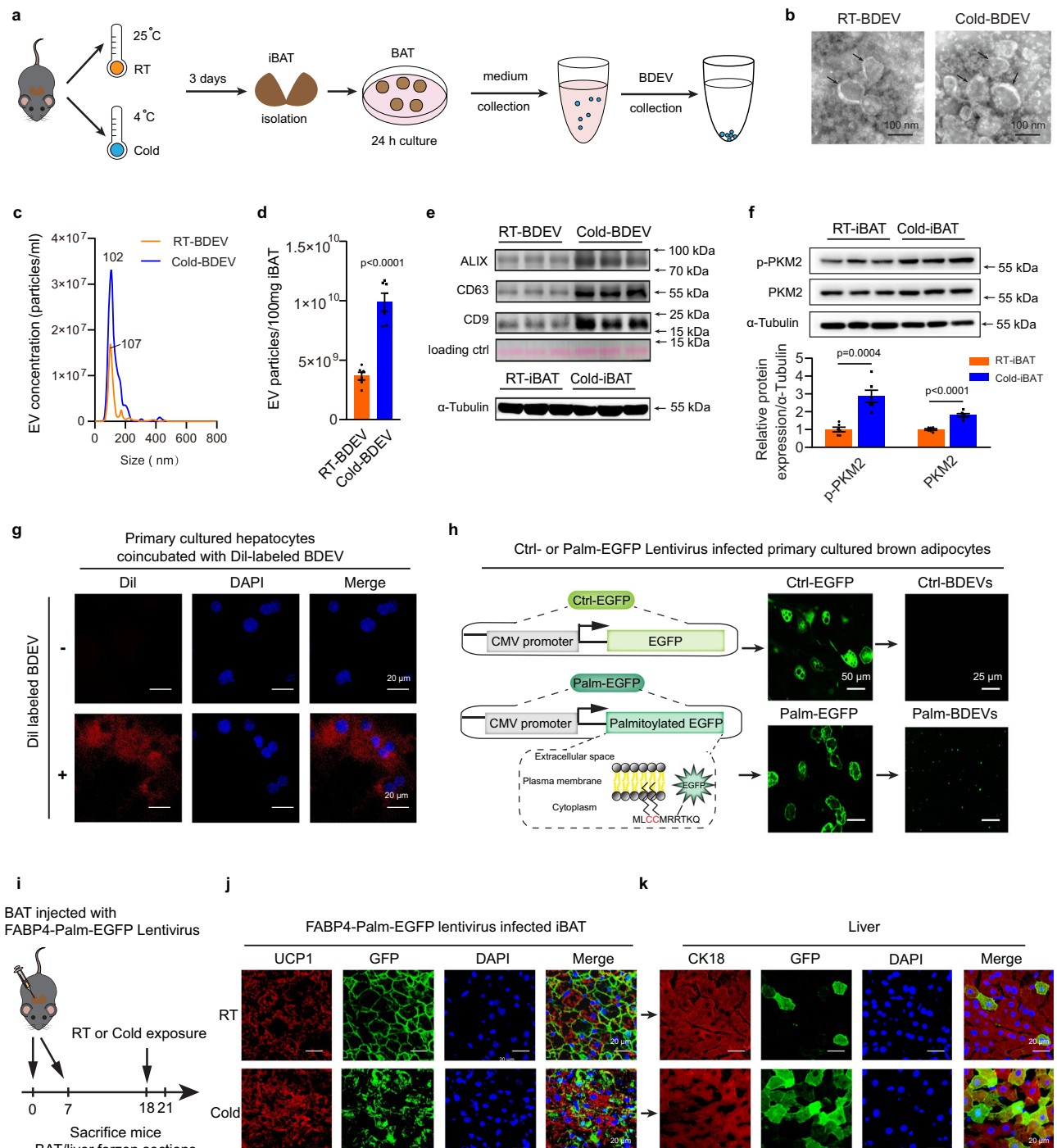

**Fig. 2 | Cold exposure induces BAT-derived EV secretion and delivery into the liver. a–e** iBAT-derived extracellular vesicle (BDEV) collection. Eight-week-old male C57BL6/J mice placed in RT (25 °C) or cold (4 °C) for 72 h. 100 mg RT-BAT or Cold-BAT was cultured for EV collection. **a** Experimental schematic. **b** Representative transmission electron microscopy (TEM) image. Scale bar: 100 nm. (*n* = 3 each group, 5 random fields per sample, representative images were shown).
**c** Nanoparticle tracking analysis (NTA) of RT-BDEVs and Cold-BDEVs (*n* = 6 each group, from 2 independent experiments). **d** Quantification of RT-BDEVs and Cold-BDEVs from 100 mg iBAT (*n* = 6 each group, from 2 independent experiments).
**e** Western blotting detection of EV markers ALIX, CD63 and CD9 in the RT-BDEVs and Cold-BDEVs. (*n* = 6 each group, from 2 independent experiments). **f** Western blotting and densitometry of total and phosphorylated PKM2 in the RT-iBAT and Cold-iBAT (*n* = 6 each group, from 2 independent experiments). **g** Confocal microscopy image of primary cultured hepatocytes co-incubation with fluorescent DiI-labelled BDEVs for 6 h. Scale bar: 20 μm. (*n* = 3 each group, 5 random fields per

sample, representative images were shown). **h** Schematic diagram of the EGFP expression reporters used. Palmitoylated GFP (Palm-EGFP) was used for EV membrane labelling (lower left panel). Microscopy images of primary cultured brown adipocytes infected with lentivirus expressing Ctrl-EGFP or Palm-EGFP reporters (medium panel, scale bar: 50 μm) and their derived BDEVs (right panel, scale bar: 25 μm). (*n* = 3 each group, 5 random fields per sample, representative images were shown). **i–k** In vivo FABP4-palm-EGFP-labelled BDEV tracking into the liver. Using multipoint injection, 1×10[8] lentiviral transduction vectors (TUs) expressing Palm-EGFP were inoculated into iBAT, and the mice were housed at RT or cold-exposed for 72 hours prior to sacrifice. **i** Experimental schematic. **j**, **k** Representative confocal microscopy image of co-immunostaining and GFP signal (green) in the iBAT (UCP1, red) **j** and the liver (CK18, red) **k** (*n* = 3 each group, 5 random fields per sample, representative images were shown). Scale bar: 20 μm. Data presented as mean ± sem. Statistical analysis was performed using two-sided unpaired t-test **d**, **f**. Source data are provided as a Source Data file.

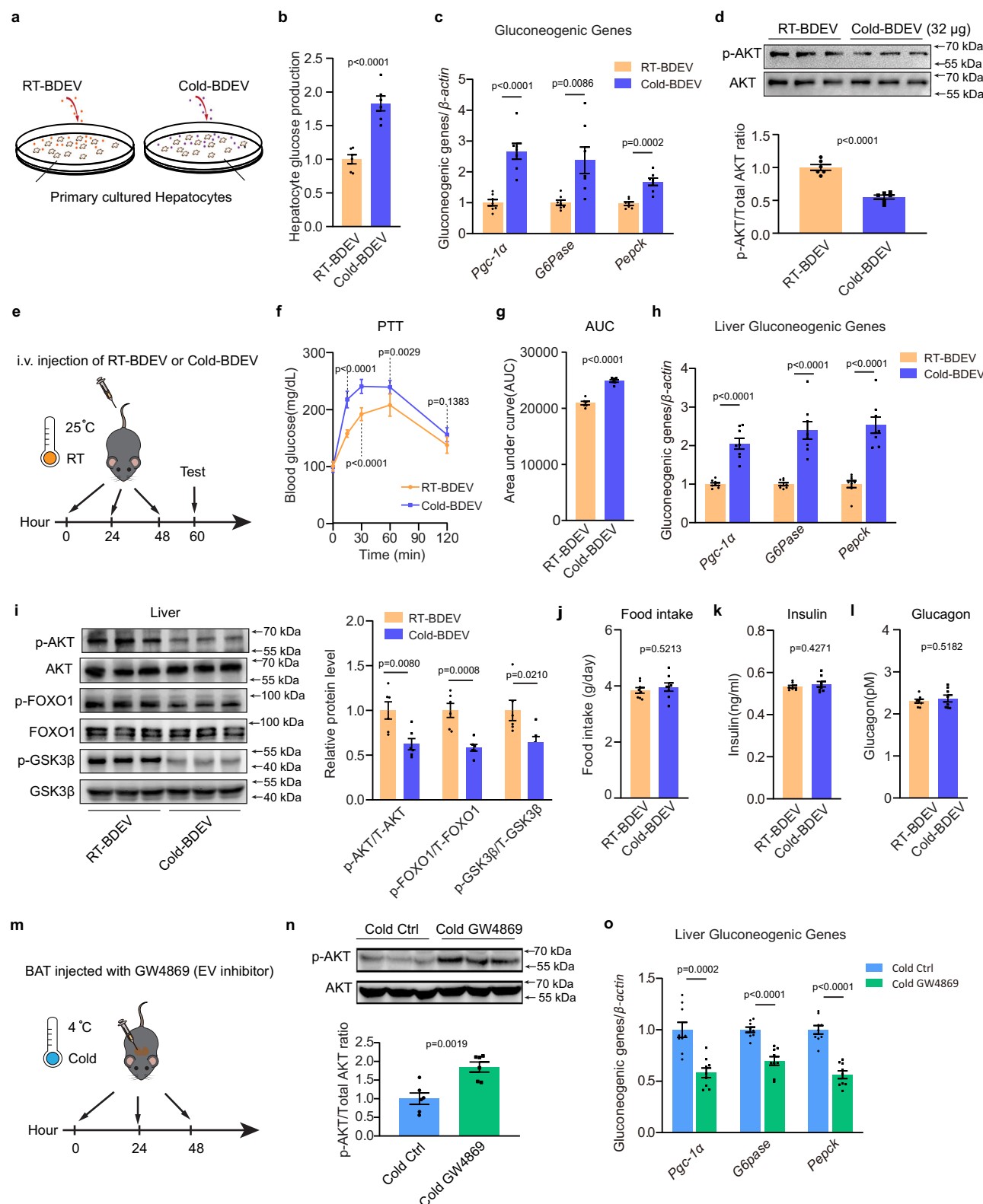

or glucagon levels (Supplementary Figure S5a-S5c). However, increased phosphorylated AKT and decreased gluconeogenic genes were observed in the mice liver after GW4869 treatment (Fig. 3n, o). These results provide compelling evidence of the critical role of activated BDEVs in promoting hepatic gluconeogenesis during cold exposure. As Cold-BDEVs fluorescent signals were also detected in WAT (Supplementary Figure S3c), their influence on lipolysis were assessed. Compared with mice treated with RT-BDEVs, Cold-BDEV

(80 µg each day) treatment resulted in elevated mRNA levels of fatty acid oxidation genes *Cpt1a, Cd36* and *Acsl1*, suggesting that Cold-BDEVs may also play a role in facilitating WAT lipolysis (Supplementary Figure S4g).

## Cold exposure induces miR-378a-3p enrichment in BDEVs
Accumulating evidence suggests that cell-derived EVs contain numerous miRNAs, and these miRNAs packaged in EVs can be taken up

**Fig. 3 | Activated BAT-derived EVs (Cold-BDEVs) enhances hepatic gluconeogenesis. a**–**d** $4 \times 10^6$ primary hepatocytes were cocultured with equal proteins of RT-BDEVs or Cold-BDEVs (32 μg) for 48 hours. **a** Schematic diagram. **b** Glucose output assay of hepatocytes ($n = 6$ biological replicates, from 2 independent experiments). **c** Relative mRNA expression of gluconeogenic genes in hepatocytes ($n = 7$ biological replicates, from 2 independent experiments). **d** Western blotting and densitometry analysis of phosphorylated and total AKT in hepatocytes ($n = 6$ biological replicates, from 2 independent experiments). **e**–**l** Eight-week-old male C57BL6/J mice were administered 80 μg equal protein of RT-BDEVs or Cold-BDEVs once daily for a total of 3 injections, and subsequent studies were performed 12 hours after the last injection. The mice were fasted for 4 h before sacrifice. **e** Schematic diagram. **f** Glucose levels during i.p. pyruvate tolerance tests (PTTs) ($n = 6$ each group, from 2 independent experiment). **g** Area under the curve of PTT, as indicated in Fig. 3f ($n = 6$ each group, from 2 independent experiment). **h** Relative mRNA expression of gluconeogenic genes in the liver ($n = 8$ each group, from 2

independent experiments). **i** Western blotting and densitometry analysis of phosphorylated and total AKT, FOXO1 and GSK3β in the liver. ($n = 6$ each group, from 2 independent experiment). Food intake **j**, serum insulin **k** and serum glucagon **l** levels in BDEV treated mice ($n = 8$ each group, from 2 independent experiments). **m**–**o** Cold-exposed eight-week-old male C57BL6/J mice were injected in situ with either GW4869 or control vehicle at 0, 24, and 48 h (for a total of 3 injections) and subsequent studies were performed 4 h after the last injection. The mice were fasted for 4 h before sacrifice. **m** Schematic diagram. **n** Western blotting and densitometry analysis of phosphorylated and total AKT in the liver ($n = 6$ each group, from 2 independent experiments). **o** Relative mRNA expression of gluconeogenic genes in the liver ($n = 9$ each group, from 3 independent experiments). Data presented as mean ± sem. Statistical analysis was performed using two-way ANOVA followed by Bonferroni's multiple comparisons test **f** and others performed two-sided unpaired t-test. Source data are provided as a Source Data file.

into recipient cells to modulate their functions. According to previous results, we wondered if BDEVs could also deliver functional miRNAs that contribute to cold-induced hepatic gluconeogenesis. We reasoned that the ideal functional BDEV-miRNA would be *(i)* expressed by BAT at moderate or high levels and *(ii)* increased in BAT, BDEVs, and serum EVs upon cold exposure. First, by filtering out miRNAs *(i)* expressed at a relatively high level in BAT via deep sequencing, miR-378a-3p and miR-193b-3p, two miRNAs that are enriched in the interscapular BAT (iBAT), showed much higher expression levels in iBAT than in other tissues, including the liver (Fig. 4a-c). According to previously published miRNA expression profiling data (GSE 41306)[35] and qPCR verification, miR-378a-3p and miR-193b-3p were selected as candidate miRNAs, as both of their expression levels were significantly increased after cold exposure (Fig. 4d, e). Notably, compared with the control group of mice, only miR-378a-3p was significantly upregulated in both BDEVs and serum EVs from cold-exposed mice, suggesting that miRNAs were selectively packaged in activated BDEVs and that the miRNAs may be selectively packaged in BDEVs under cold stress (Fig. 4f, g and Supplementary Figure S6a-S6c). Thus, miR-378a-3p was chosen for further investigation. As shown in Fig. 4h, the expression level of miR-378a-3p was markedly increased in the livers of cold-exposed mice. To examine whether the increased miR-378a-3p level resulted from higher hepatic transcription levels, we measured primary miR-378a expression in iBAT and the liver. As shown in Fig. 4i, the transcriptional level of miR-378a (primary miR-378a) was significantly increased in iBAT but not in the liver after cold exposure. To study the possible reason leading to distinct responses of miR-378a-3p transcription in BAT and liver to cold, the expression of *Ffar4*, which has been reported to positively regulate miR-378a in the BAT, was assessed. Cold exposure significantly increased the *Ffar4* mRNA level in BAT, but decreased its expression in the liver, which was consistent with the transcription level of miR-378a-3p (Supplementary Figure S6d and Fig. 4i). Considering that the expression of *Ffar4* in the liver was much lower in the liver than in BAT, *Lxrα* (Liver X receptor α), another reported transcription activator of miR-378a in the liver[36], was also examined. Compared with the control group, the expression of *Lxrα* was markedly repressed in the liver from cold-exposed mice (Supplementary Figure S6e). These data suggested that in response to cold stress, the transcription of miR-378a-3p in the BAT and liver may be differentially regulated. Next, 80 μg RT-BDEVs (~0.49 fmol miR-378a /μg RT-BDEV) and Cold-BDEVs (~1.75 fmol miR-378a /μg Cold-BDEV) were intravenously administered to mice each day for 3 days (Supplementary Figure S6f). The expression of miR-378a-3p was ~79 fmol/g liver tissue before treatment, after injection, miR-378a-3p level increased to ~102 fmol/g liver in RT-BDEV treated mice and to ~182 fmol/g liver in Cold-BDEV treated mice (Fig. 4j), respectively. In addition, local injection of GW4869 (inhibit EV release) in BAT led to significantly decreased miR-378a-3p level in the liver of cold exposed mice but not fasted mice (Fig. 4k, l, Supplementary Figure S6i-S6k).

The observed differences in cold-exposed and fasted mice were reasonable, as the increased miR-378a-3p expression was resulted from the elevated transcription of miR-378a in the liver from fasted mice (Supplementary Figure S6g-S6h), whereas for cold-exposed mice, the increased miR-378a-3p was not from its own transcription activation in the liver (Fig. 4i, m). In addition, the expression of Dicer, which directly regulates miRNA maturation, was unchanged in the liver of cold-exposed mice (Supplementary Figure S6l-S6m). Moreover, the uneven changes in several other miRNA expression levels in the cold-exposed liver, including miR-21, -192, and -223, suggested that the increased miR-378a-3p was unlikely due to the change of the miRNA processing proteins in the liver (Supplementary Figure S6n). Furthermore, compared with the control group, mice with BATectomy exhibited markedly lower expression levels of miR-378a-3p in the serum EVs and liver after cold exposure, without a significant change in primary miR-378a in the liver (Fig. 4n–p). These results strongly indicate that the upregulated hepatic miR-378a-3p was likely not from hepatic de novo synthesis but delivery of the BDEV-enriched miR-378a-3p into the liver upon cold exposure.

## BDEV-derived miR-378a-3p promotes gluconeogenesis

Next, we investigated the possible role of miR-378a-3p in regulating hepatocyte glucose metabolism. First, primary cultured brown adipocytes were transfected with a fluorescent Cy3-labelled miR-378a-3p mimic, and BDEVs were collected and incubated with hepatocytes. The appearance of red fluorescent Cy3 dye in the hepatocytes demonstrated that the Cy3-miR-378a-3p mimic could be directly delivered from brown adipocytes to recipient hepatocytes via BDEVs (Fig. 5a). To assess the biological function of miR-378a-3p in hepatocytes, scrambled nucleotide (Ctrl-NC) or miR-378a-3p was overexpressed in hepatocytes. Compared with control cells, overexpression of miR-378a-3p significantly induced glucose production in cultured hepatocytes, concurrent with upregulated gluconeogenic gene expression and reduced protein levels of p-AKT, p-FOXO1 and p-GSK3β in hepatocytes (Fig. 5b–d). These results suggested that BDEV-enriched miR-378a-3p might be a functional signalling molecule in regulating hepatocyte gluconeogenesis. p110α, the fundamental subunit of phosphoinositide 3-kinase (PI3K) that controls glucose metabolism, was demonstrated to be a direct target of miR-378a-3p in the previous study[37]. Thus, the protein level of p110α was examined in the hepatocytes. As revealed by western blotting, overexpression of miR-378a-3p did not significantly change the phosphorylation level of IR or total IR but markedly decreased the p110α protein level in hepatocytes (Fig. 5e). To explore the influence of Cold-BDEV-derived miR-378a-3p on hepatocytes, primary hepatocytes were transfected with different doses of anti-miR-378a (0 nM, 10 nM, 25 nM, 50 nM, 100 nM, and 200 nM) and co-incubated with Cold-BDEVs for 48 h (Fig. 5f). When the doses of the anti-miR-378a gradually increased, the protein level of p110α was also elevated in a dose-dependent manner, and reached the maximal level

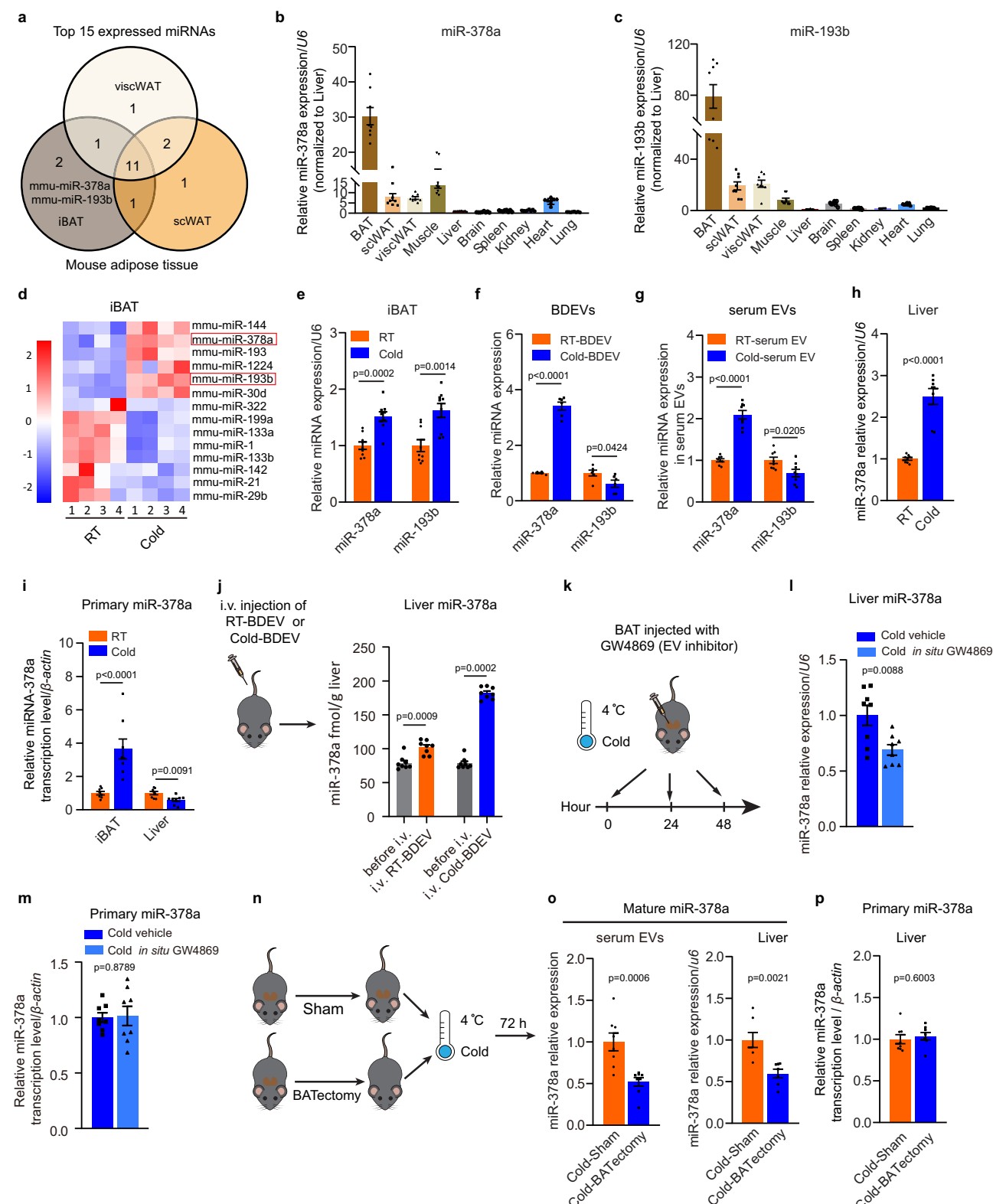

at 100 nM treatment and maintained thereafter (even 200 nM treatment could not cause further induction, Fig. 5g, h). In line with the gradually increased protein levels of p110α, the expression of the gluconeogenic genes was downregulated in a dose-dependent manner and decreased to the lowest level at 100 nM treatment (Fig. 5i). To further determine the possible contribution of miR-378a-3p to Cold-BDEVs, the hepatocytes co-incubated with Cold-BDEVs were treated with/out 100 nM anti-miR-378a-3p. In control hepatocytes untreated

with anti-miR-378a, Cold-BDEVs administration significantly reduced p110α protein level and markedly increased gluconeogenic gene expressions (~2-fold increase) (Supplementary Figure S7a-S7b). While in those cells treated with anti-miR-378a (100 nM), comparable protein levels of p110α were detected in hepatocytes treated with or without Cold-BDEVs, suggesting that the function of miR-378a-3p that derived from Cold-BDEVs were completely abolished in hepatocytes (Supplementary Figure S7c). Notably, only slight increases (~1.2-fold)

**Fig. 4 | Cold exposure increases miR-378a-3p expression in BDEVs. a** Venn diagram depicting the number fat depot-specific miRNAs within the top 15 expressed miRNAs in each indicated fat depot (*n* = 6 each group, mixed samples). **b, c** qPCR analysis of the relative levels of miR-378a-3p **b** and miR-193b-3p **c** (*n* = 8 each group, from 2 independent experiments). **d** Heatmap showing Z scores of miRNA expression in iBAT of mice placed at RT or cold for 72 h (GSE 41306[35], *n* = 4 per group). **e**–**i** Eight-week-old male C57BL6/J mice were placed in RT or cold for 72 hours. **e**–**g** qPCR analysis of the levels of miR-378a-3p and miR-193b-3p in iBAT (**e**, *n* = 9 each group, from 2 independent experiments); in the equal amounts of RT-BDEVs or Cold-BDEVs (**f**, *n* = 6 each group, from 2 independent experiments) and serum-EVs from the same volume of serum (**g**, *n* = 8 each group, from 2 independent experiments). **h** qPCR analysis of the relative level of miR-378a-3p in liver (*n* = 8 each group, from 2 independent experiments). **i** qPCR analysis of the relative level of primary miR-378a-3p in iBAT and liver (*n* = 8 each group, from 2 independent

experiments). **j** Eight-week-old male C57BL6/J mice treated with BDEVs each day for 3 injections (left panel). qPCR analysis of the absolute level of miR-378a in the liver 12 h post last injection. (*n* = 8 each group, from 2 independent experiments). **k**–**m** Cold-exposed eight-week-old male C57BL6/J mice were injected in situ with either GW4869 or control vehicle. Schematic diagram **k**; qPCR analysis of the relative miR-378a-3p level **l** and primary miR-378a-3p level **m** in liver. (*n* = 8 each group, from 2 independent experiments) **n**–**p** Sham-operated mice or mice with BATectomy were cold-exposed for 72 h. Experimental schematic **n**; qPCR analysis of the relative miR-378a-3p levels in the serum-EVs and liver **o** and primary miR-378a-3p in liver **p** (*n* = 8 each group, from 2 independent group). Mice were fasted for 4 h before sacrifice. Data presented as mean ± sem. Statistical analysis was performed using Mann Whitney test (two tailed) (**j**, **o** left panel) and others performed two-sided unpaired t-test. Source data are provided as a Source Data file.

expressions of gluconeogenic genes were observed (Supplementary Figure S7d). These results suggested that as a cargo RNA in Cold-BDEVs, miR-378a-3p plays an important role in promoting hepatic gluconeogenesis Fig. 5j.

Given that miR-378a-3p increased gluconeogenesis in primary cultured hepatocytes, we evaluated the effects of BDEV-contained miR-378a-3p in an in vivo context. As shown in Fig. 6a, to achieve miR-378a-3p overexpression in the iBAT but not in the liver, the adeno-associated virus (AAV) expressing miR-378a-3p was constructed under the adipose tissue-specific promoter FABP4 (AAV$^{FABP4}$-378a) and introduced into the iBAT of mice via multipoint injection. At 28 days post-infection, the expression of miR-378a-3p was significantly increased in iBAT (Fig. 6b). Compared with the control group, the miR-378a-3p expression level was also increased in both the serum EVs and livers of the mice infected with AAV$^{FABP4}$-378a, indicating that overexpression of miR-378a-3p in iBAT increased miR-378a-3p secretion into the circulation and delivery into the liver (Fig. 6c, d). Notably, the introduction of miR-378a-3p into iBAT did not alter the endogenous miR-378a-3p transcription level in the iBAT or the liver (Fig. 6e). There were no overt differences in body weight, food intake, or insulin and glucagon levels between the mice expressing miR-378a-3p and the control animals, while the blood glucose was slightly increased (Fig. 6f–j). Notably, increased hepatic gluconeogenesis was observed in miR-378a-3p-overexpressing mice, as indicated by the enhanced expression of gluconeogenic genes and decreased p-AKT and p110α signalling in the liver of AAV$^{FABP4}$-378a-infected mice, while the phosphorylation and total levels of IR were unchanged, confirming that miR-378a-3p directly affects factors downstream of IR by targeting p110α in the liver (Fig. 6k–m). Taken together, these data confirmed that BDEV-contained miR-378a-3p can promote gluconeogenesis by directly targeting hepatic p110α in vivo.

### Restoration of miR-378a-3p in iBAT rescues hepatic gluconeogenesis in miR-378 KO mice
To investigate the role of miR-378a-3p in regulating glucose homeostasis during cold exposure, we generated a miR-378a KO mouse model (378KO) using the CRISPR/Cas9 system (Fig. 7a, b). The characterization of 378KO mice is shown in Supplementary Figure S8a-S8e. As predicted from the above findings, increased protein level of p110α was detected in the liver of 378KO mice housed at RT (Fig. 7c). No significant alteration in body weight or food intake was detected, while a slightly decreased blood glucose level was observed in 378KO mice (Fig. 7d–f). There were no overt differences in insulin and glucagon levels between the 378KO mice and the control animals (Supplementary Figure S8f-S8g). Because miR-378a is located in the first intron of *Pgc-1β*, we also examined whether PGC-1β was influenced in 378KO mice. As shown in Fig. 7g, the depletion of miR-378a did not alter the expression of PGC-1β in the livers of mice. The hepatic miR-378 level was upregulated in the cold-exposed WT mice, concurrent with decreased p110α protein level in the liver. We examined the effect of

cold exposure on 378 KO mice. As shown in Fig. 7h, i, compared with cold-exposed WT controls, loss of miR-378a showed a markedly increased protein level of p110α. Consistently, significantly increased AKT phosphorylation and decreased expression of *Pgc-1α*, *G6Pase,* and *Pepck* were observed in the liver of cold-exposed 378 KO mice (Fig. 7j–k), indicating the impaired cold-induced hepatic gluconeogenesis. Not surprisingly, we also observed decreased blood glucose levels in cold-exposed 378 KO mice (Fig. 7l). Moreover, to directly explore the function of BAT-derived miR-378a-3p in regulating hepatic gluconeogenesis during cold exposure in KO mice, we took advantage of AAV$^{FABP4}$-378a to restore miR-378a-3p in the iBAT by multipoint injection (Fig. 7m, n). As shown in Fig. 7o, re-expression of miR-378a-3p in iBAT significantly elevated *G6Pase* and *Pepck* in the liver of 378a KO mice, further confirming that BAT-derived miR-378a-3p directly promotes hepatic gluconeogenesis.

### Depletion of miR-378a in BAT impairs cold-induced hepatic gluconeogenesis
To specifically knockdown miR-378a-3p in the liver, AAV expressing a sequence encoding a string of antisense sequences to miR-378a-3p (AAV$^{TBG}$-sponge) was constructed under the liver-specific promoter TBG and i.v. injected into mice and housed at RT. As schematically illustrated in Fig. 8a, 25 days post-infection, mice were cold-exposed for 72 h. Knockdown of miR-378a-3p in the liver resulted in significantly increased protein levels of p110α and p-AKT (Fig. 8b), along with decreased expressions of *G6Pase* and *Pepck, Pgc-1α* (Fig. 8c). These data confirmed that depletion of miR-378a-3p in the liver impairs cold-induced gluconeogenesis.

To directly demonstrate whether specific inhibition of miR-378a-3p in iBAT could influence cold-induced hepatic gluconeogenesis. AAV$^{FABP4}$-sponge was constructed and locally injected into the iBAT. 25 days post infection, mice were cold-exposed for 3 days prior to sacrifice (Fig. 8d). Compared with the control group (AAV$^{FABP4}$-NC), miR-378a-3p was decreased in BDEVs, serum-EVs and the liver of mice infected with AAV$^{FABP4}$-sponge, while the transcription of miR-378a-3p in the liver was unaffected, indicating that binding of miR-378a-3p in BAT resulted in reduced EV-mediated miR-378-3p secretion into the liver (Fig. 8e–h). Inhibition of miR-378a in iBAT did not alter the body weight, food intake or insulin level in cold-exposed mice (Fig. 8i–k). Knockdown of miR-378a in iBAT significantly increased the protein levels of p110α and p-AKT in the liver, and repressed hepatic gluconeogenic gene expression (Fig. 8l–n), thereby providing direct evidence that inhibition of miR-378a in BAT could suppress cold-induced hepatic gluconeogenesis (Fig. 8o).

## Discussion
Maintaining blood glucose concentrations in a constant homoeostatic range under different physiological conditions is essential to the survival of mammals. BAT, which oxidizes metabolites to produce heat, is highly active during cold exposure and is essential in regulating whole-

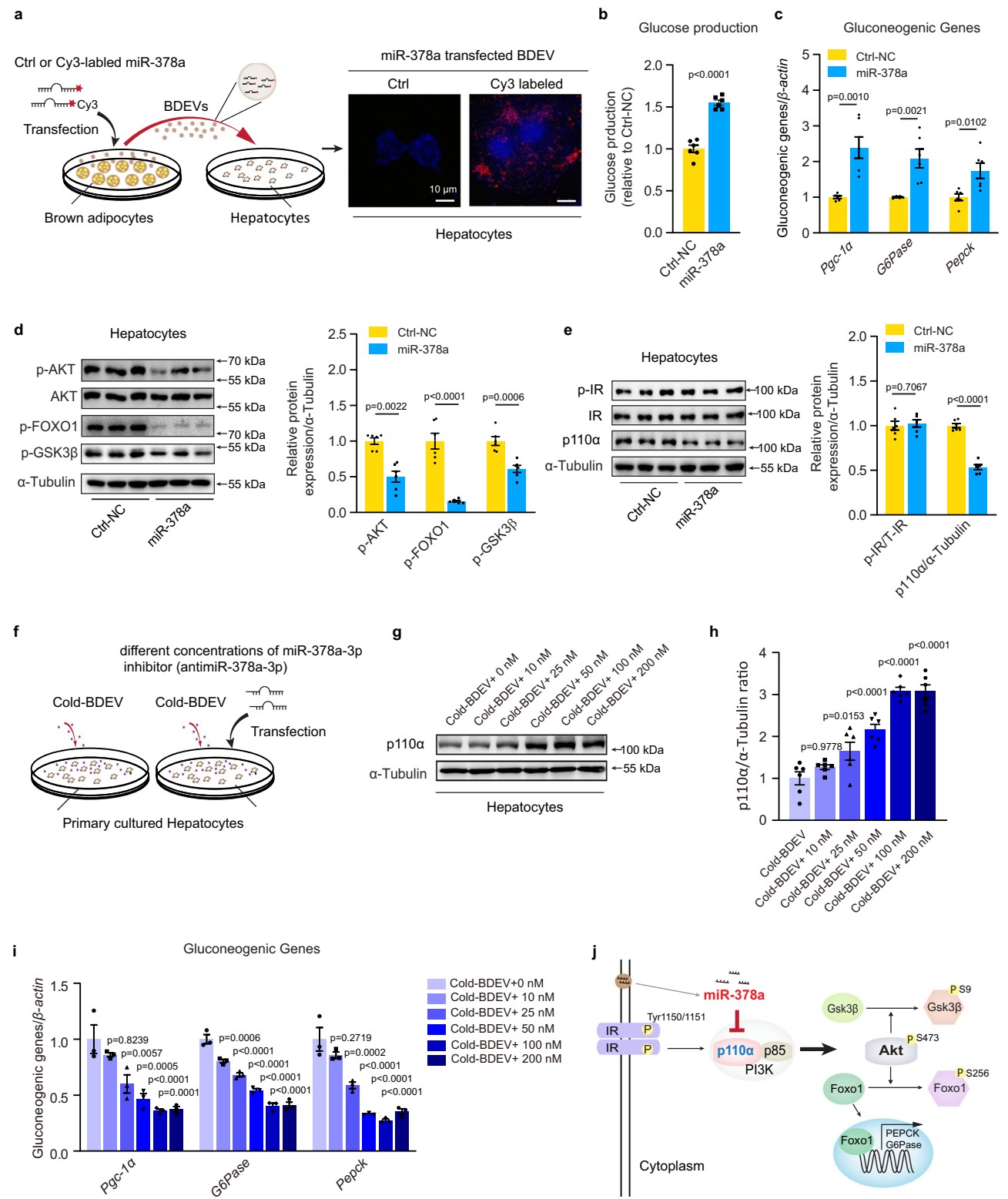

body energy balance. Recent studies have emphasized that BAT activation markedly increases glucose uptake from the circulation and that coordination of organ-specific responses may be required to maintain blood glucose homoeostasis[6,21]. Beyond its thermogenic ability, BAT has recently been recognized as an important endocrine organ. However, unlike the classical WAT adipokines, much less is known about the extent to which BAT engages other organs via its release of "batokines", especially under cold stress. The present study shows that EV-containing miR-378a-3p secreted by BAT stimulates

hepatic glucose production to maintain glucose homoeostasis under cold stress. In this circulating miR-378a-3p-based mechanism, cold-activated BAT releases miR-378a-3p into the circulation via an EV-dependent pathway, and miR-378a-3p plays a role in facilitating hepatic gluconeogenesis to maintain systemic glucose balance.

Of the few consistent studies pointing to a role of BAT secretome on hepatic gluconeogenesis, Qing et al. reported that acute psychological stress induced BAT-dependent IL-6 secretion, thereby mediating hyperglycaemia through hepatic gluconeogenesis, which is

**Fig. 5 | Cold-BDEV-derived miR-378a-3p enhances gluconeogenesis in cultured hepatocytes by targeting p110α. a** Experimental procedure for primary cultured hepatocytes cocultured with miR-378a-3p-overexpressing BDEVs. Primary cultured brown adipocytes were transfected with a Cy3-labelled miR-378a-3p mimic, the EVs secreted by adipocytes were cocultured with primary cultured hepatocytes (left panel). Images of hepatocytes incubated with BDEVs bearing Cy3-tagged miR-378a-3p for 6 hours. Untreated hepatocytes were used as a negative control (right panel). Scale bar: 10 μm. **b**–**e** Primary cultured hepatocytes were transfected with miR-378a-3p for 48 h. Hepatocytes treated with scrambled nucleotide were used as a control. (*n* = 6 biologically replicates for each group, from 2 independent experiment). **b** The glucose output assay of administered hepatocytes. **c** qPCR analysis of the relative expression levels of gluconeogenic genes in hepatocytes. **d** Western blotting and densitometry analysis of p-AKT, p-FOXO1, and p-GSK3β in hepatocytes. α-Tubulin was used as an internal control. **e** Western blotting and densitometry analysis of phosphorylated IR, total IR, and p110α in hepatocytes transfected with NC nucleotides or miR-378a-3p mimic for 48 hours. **f**–**i** $4 \times 10^6$ primary cultured hepatocytes transfected with different doses of anti-miR-378a (0 nM, 10 nM, 25 nM, 50 nM, 100 nM, 200 nM) and co-incubation with 32 μg Cold-BDEVs for 48 h. **f** Schematic diagram. **g**, **h** Western blotting detection and densitometry analysis of p110α in hepatocytes. (*n* = 6 each group, from 2 independent experiment). **i** Relative mRNA expression of gluconeogenic genes in hepatocytes (*n* = 3 biological replicates). **j** Graphic illustration of the miR-378a-3p targeting pathway in hepatocytes. Statistical analysis was performed using one-way ANOVA followed by Bonferroni's multiple comparisons test **h**, **i** and others performed two-sided unpaired t-test. Source data are provided as a Source Data file.

necessary for fuelling "fight or flight" responses and independently of thermoregulation[38]. In the present study, we showed that upon cold exposure, in addition to the well-determined thermogenic role, BAT also functions as a stress-responsive endocrine organ by secreting "RNAkines".Although several studies have indicated that adrenergic stimulation can activate hepatic gluconeogenesis through the cAMP-PKA pathway during cold acclimation[39], our results provide evidence that EV-mediated BAT-liver crosstalk also contributes to this physiological process to maintain whole-body glucose balance. Of note, our results indicate that Cold-BDEVs may not only stimulates hepatic gluconeogenesis for glucose replenishment, but also play roles in regulating WAT lipolysis for fatty acid supply, which may deserve further exploration.

Accumulating evidence suggests EVs' critical role in delivering messages between different cells and organs. Specifically, adipose tissue-derived miRNAs in EVs can sail through the blood circulation and reach distant organs to regulate gene expression. Thomou et al. reported that EV-miRNAs secreted by BAT can regulate gene expression in other tissues, confirming that BAT-derived EVs are newly identified endocrine mediators in controlling whole-body energy homoeostasis[31]. Nevertheless, questions remain. BAT is highly activated during cold exposure. Recently, Sugimoto et al. reported that BAT-derived MaR2 contributes to the cold-induced resolution of inflammation[21]. Here, we asked whether the activation of BAT influences EV release and function. Analysis of activated BAT-derived EVs' qualitative and quantitative characteristics is expected to reveal important insights into their endocrine functions. We found that PKM2, involved in glycolysis and tumour EV release[34], also promoted BAT-derived EV secretion under cold stress. However, a standardized methodological approach for BAT-derived EVs has yet to be fully established. The level of UCP1, the most important thermogenic protein in BAT, was reported to be significantly up-regulated in obese BAT-derived EVs in a previous study[40]. Moreover, another study demonstrated that adrenergically stressed brown adipocytes released damaged mitochondrial components through large EV fractions, but not in exosomes with size ~50 nm[41]. In the present study, we found that for mice housed at RT, UCP1 protein was not enriched in the small-sized BAT-EVs (BDEVs, ~100 nm) compared to medium/large EV fractions (>200 nm). We speculated that the observed differences may be due to variations in animal models, metabolic conditions, EV extraction methods, EV protein selection, and comparison methods, testing of specific EV subpopulations, and the amount of EVs used for immunoblot detection in different studies.

Of note, although two miRNAs−miR-378a-3p and miR-193b-3p, more highly expressed in BAT than in other tissues- were examined in Cold-BDEV, only miR-378a-3p was increased in cold-activated BAT and BDEVs, suggesting that the miRNAs were selectively packaged into the BDEVs during cold exposure. Moreover, this lack of concordance between the activation-regulated miRNAs in BAT and its accumulation in BDEVs indicates that miRNA loading into EVs involves an active selection rather than a passive process. This observation reminds us of previous studies reporting that the miRNA repertoires of EVs were different from those of their parental cells[42]. However, although miRNA biogenesis has been extensively studied, little is known about the selection of miRNAs selectively loaded into EVs under different physiological and pathological conditions. Garcia-Martin et al. proposed that sequence motifs present in miRNAs could control their loading into EVs[43]. In agreement with their results, the short motif "GGAG", which is present in miRNAs that tend to be localized in EVs, was found in the mature sequence of miR-378a-3p. Nevertheless, how miRNAs are selectively loaded into EVs and exported from activated BAT requires further study and exploration.

Although adipose tissue (AT)-derived EV-miRNAs have been studied in several pathological conditions, such as obesity and diabetes[44], little is known about the consequences of circulating miRNAs from AT under physiological conditions. The present study demonstrated that BDEV-enriched miR-378a-3p secreted by cold-activated brown fat stimulates hepatic glucose production to maintain systemic glucose homoeostasis. By identifying the role of EV-miR-378a-3p in regulating hepatic glucose metabolism under cold stress, our findings highlight the cold-responsive endocrine function of BAT.

## Methods
### Animals
All animal experimental procedures were conducted in accordance with the National Institutes of Health Guide for the Care and Use of Laboratory Animals and were approved by the Animal Ethical Board of Nanjing University (IACUC-2104002). To reduce the variations that may result from hormonal changes and avoid the possible influences caused by the estrogen cycle on glucose metabolism in female mice[45–47], only male C57BL/6 mice were used in the present study. Eight-week-old male C57BL/6 J WT mice were from GemPharmatech Laboratory (Nanjing, China). The miR-378 KO mice were generated on a C57BL/6 J background at the GemPharmatech Laboratory (Nanjing, China). We used CRISPR/Cas9 technology to modify miR-378a gene, exon1 of miR-378a-201 (ENSMUST00000198300.1) transcript is recommended as the knockout region. Primers used for genotyping are shown in Supplementary Table 1. Heterozygous 378KO mice were mated to produce littermate WT and KO mice for study. To calculate food intake, the experimental animals were raised individually in a specific pathogen free (SPF) facility at Nanjing University and maintained at $21 \pm 2\,^{\circ}C$ and a relative humidity of $55 \pm 10\%$ with free access to pellet normal chow diet (cat#SWS9102, XieTong Biology) and water and kept on a 12 h light/12 h dark cycle. For cold exposure, the mice were housed individually in a 4°C incubator for 72 h, and the mice were weighed before and after cold (Δ body weight). RT or cold exposure mice were fasted for four hours before sacrifice, as previously reported[38]. The mice were euthanized by carbon dioxide inhalation. Other details of in vivo experiments were described in each section.

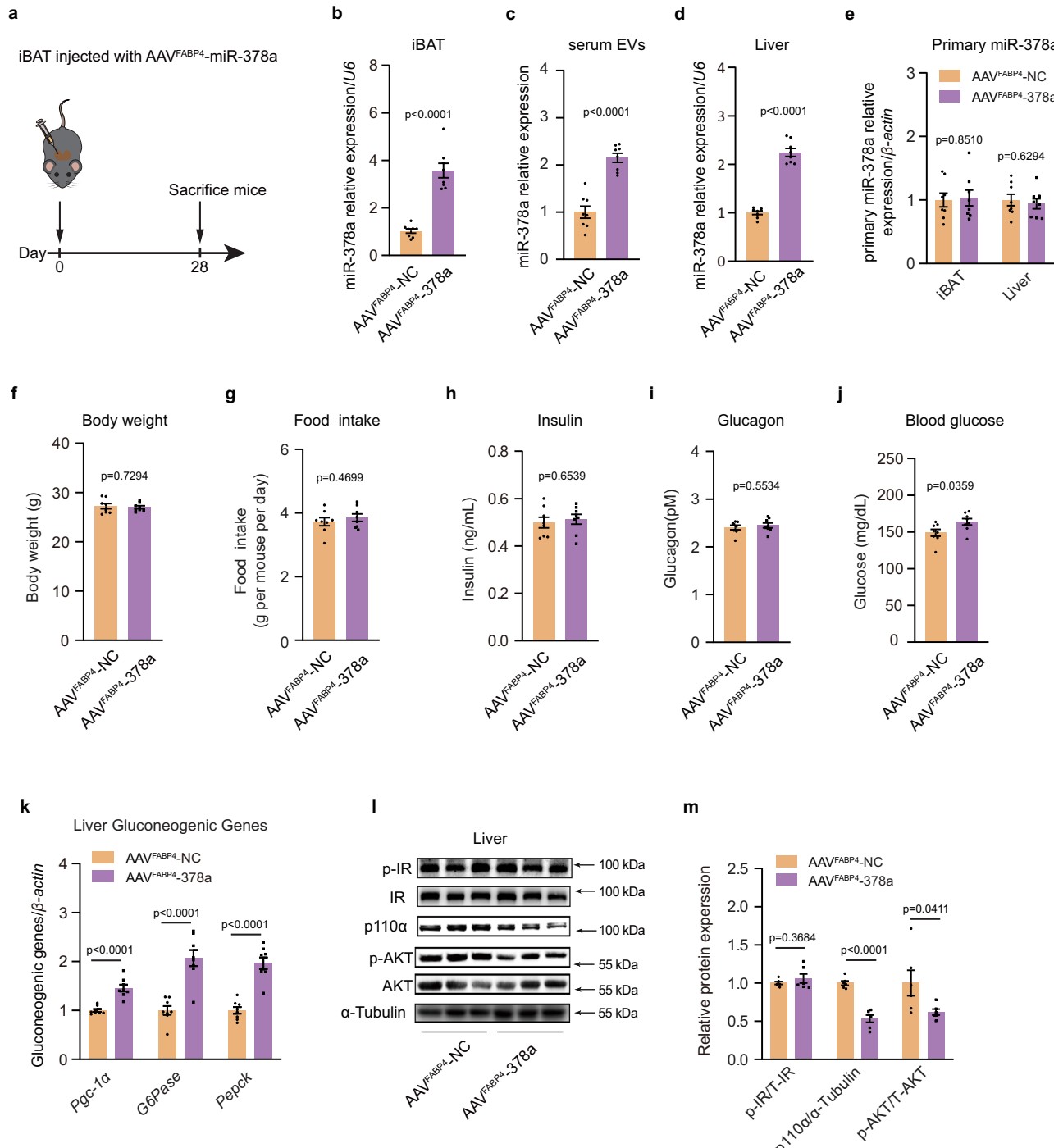

**Fig. 6 | Overexpression of miR-378a-3p in BAT regulates liver gluconeogenesis in mice. a**–**m** iBAT-specific overexpression of miR-378a-3p induced with adeno-associated virus constructed under the adipose tissue-specific promoter FABP4 (AAV$^{FABP4}$-378a), a total of 50 µl of 1 ×10$^9$ Vg/µl of each AAV was injected into the iBAT in situ of eight-week-old male C57BL6/J mice. Control mice were injected with scrambled Control (AAV$^{FABP4}$-NC). 28 days post AAV injection, mice were fasted 4 h, followed by qRT-PCR analysis of the relative miR-378a-3p expression level in **b** iBAT; **c** serum EVs; and **d** livers (*n* = 8 each group, from 2 independent experiments). **e** Relative level of primary miR-378a-3p in the iBAT and liver (*n* = 8 each group, from 2 independent experiments); **f** body weight; **g** food intake; **h** serum insulin level; **i** serum glucagon level; **j** blood glucose level; (*n* = 8 each group, from 2 independent experiments); **k** qPCR analysis of the mRNA levels of gluconeogenic genes in the liver (*n* = 8 each group, from 2 independent experiments); **l** Western blotting for p-IR, p-AKT, total IR, AKT and p110α in the liver and **m** densitometry analysis of p-IR, p-AKT, and p110α. α-Tubulin was used as an internal control (*n* = 6 each group, from 2 independent experiments). Data presented as mean ± s.e.m. All statistical analysis was performed using two-sided unpaired t-test. Source data are provided as a Source Data file.

### Primary cell isolation and culture

Primary adipocytes were isolated and cultured as previously described[48]. Briefly, the interscapular brown adipose tissue (iBAT) of 7–8-week-old male C57BL/6 J mice was obtained, minced and digested in DMEM basic (cat#C11995500BT,Gibco) containing collagenase D (10 mg/mL) (cat#11088882001, Sigma), dispase II (2.4 U/mL) (cat# 4942078001,Sigma) and CaCl2 (10 mM) (cat#20011160, Shanghai trial of Sinopharm) at 37 °C for 20 min. The tissue suspension was filtered

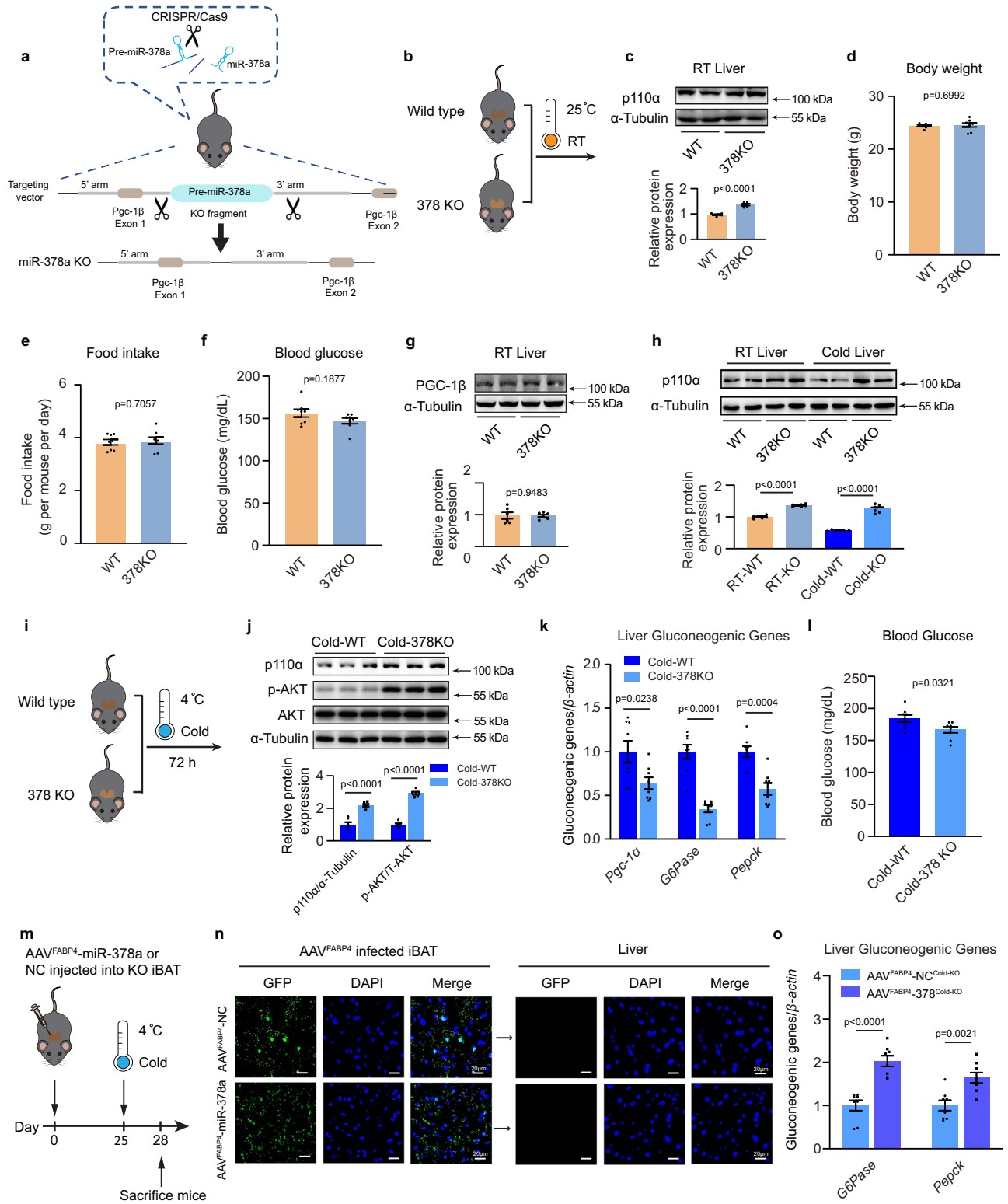

by 100 μm and 40 μm cell strainers (BD Biosciences), and cell pellets were resuspended in DMEM/F-12 (cat#C11330500BT,Invitrogen) supplemented with 10% (vol/vol) fetal bovine serum (FBS) (cat#1099141 C, Gibco)and 1% (vol/vol) penicillin/streptomycin (P/S) (cat#15140122, Gibco) and plated. After 2 days of culture, the primary pre-adipocytes were exposed to induction by DMEM/F-12 containing indomethacin (125 μM) (cat#I-7378, Sigma), dexamethasone (5 μM) (cat# D-1756, Sigma), insulin (0.5 μg/mL) (cat#I0546; Sigma), iso-butylmethylxanthine (0.5 mM) (cat# I-5879, Sigma), rosiglitazone

(1 μM) (cat# R-2408, Sigma), T3 (1 nM) (cat# T-2877, Sigma), and 10% FBS. From day 4 after induction, the cells were maintained in medium containing insulin (0.5 μg/mL), T3 (1 nM), rosiglitazone (1 μM) and 10% FBS until they were collected.

Primary hepatocytes were isolated from mice by two-step col-lagenase perfusion[49]. C57BL/6 J mice at 7-8 weeks of age were anaes-thetized by isoflurane inhalation (3–4% induction, 1–2% maintenance), after placing catheter into the portal vein, the inferior vena cava was cut and the liver was perfused at 5–7 ml/min with pre-warmed

**Fig. 7 | Restoration of miR-378a-3p in iBAT rescues hepatic gluconeogenesis in miR-378 KO mice. a** Schematic diagram of the miR-378 gene genomic locus and construction for miR-378 KO (378 KO) mice. **b**–**g** Eight-week-old male miR-378 KO mice and control WT littermates housed at RT. **b** Schematic diagram. **c** Western blotting and densitometry analysis of p110α and in liver ($n = 6$ each group, from 3 independent experiments). **d** Body weight; **e** food intake; **f** blood glucose ($n = 8$ each group, from 2 independent experiments); **g** western blotting and densitometry analysis of PGC-1β in liver ($n = 6$ each group, from 3 independent experiments). **h** Western blotting and densitometry analysis of p110α in the liver of WT and KO mice at RT or cold for 72 h ($n = 6$ each group, from 3 independent experiments). **i**–**l** Eight-week-old male miR-378 KO mice and control WT littermates were cold-exposed for 72 h. **i** Schematic illustration. **j** Western blotting and densitometry analysis of p110α, p-AKT, and total AKT in the liver ($n = 6$ each group, from 2 independent experiments). **k** qPCR analysis of gluconeogenic genes in the liver ($n = 8$ for each group, from 2 independent experiments). **l** Blood glucose in mice

following cold exposure ($n = 8$ each group, from 2 independent experiments). **m**–**o** iBAT-specific overexpression of miR-378a-3p induced with adeno-associated virus constructed under the adipose tissue-specific promoter FABP4 (AAV[FABP4]-378a), a total of 50 μl of $1 \times 10^9$ Vg/μl of each AAV was injected into the iBAT in situ of eight-week-old male 378KO mice. Control mice were injected with AAV[FABP4]-NC. **m** Schematic illustration. **n** Representative confocal microscopy image of GFP signal (green) in the iBAT (left panel) and the liver (right panel) of mice infected with AAV ($n = 3$ each group, 5 random fields per sample, representative images were shown). Scale bar: 20 μm. **o** qRT-PCR analysis of the gluconeogenic genes in the livers of cold-exposed KO mice expressing miR-378a ($n = 8$ each group, from 2 independent experiments). Mice were fasted for 4 h before sacrifice. Data presented as mean ± sem. Statistical analysis was performed using Mann Whitney test (two-tailed) **f** and others performed two-sided unpaired t-test. Source data are provided as a Source Data file.

perfusion medium (cat#17701038, Gibco) for 10 minutes. Then, perfusion was performed with pre-warmed Digestion Medium including Type IV collagenase (cat#17104019, Gibco) at 5 ml/min for 1-2 minutes. After dissociation, cells were filtered through a 70 μm filter. Hepatocytes were further separated and purified by centrifugation at low speed (450 g, 5 min), then cultured in RPMI medium 1640 (cat#C22400500BT, Gibco) supplemented with 2% FBS and 1% P/S.

### Extracellular vesicles (EV) Collection
BAT-derived extracellular vehicles (BDEVs) were isolated via ultracentrifugation. The EVs from the cultured interscapular brown adipose tissue (iBAT) were isolated as follows: the iBAT of mice was excised and transferred into 2 ml DMEM with 1% P/S and gently cut into 2 mm³ pieces. The chopped tissues were centrifuged 5 min at $1000 \times g$ at room temperature, pellets were resuspended in DMEM and transferred into a petri dish. After 30 min incubation in DMEM containing 2% EV-free FBS, 0.5 mg/ml DNase I, 0.2 mg/ml RNase A and 1% P/S, the culture medium was exchanged. The iBAT tissues were incubated for 24 h at 37 °C, 5% CO₂, and then culture supernatant was collected for EV isolation by ultracentrifugation method. In brief, the medium was centrifuged at $300 \times g$ for 10 min, $3000 \times g$ for 20 min, and $10,000 \times g$ for 30 min to remove tissues, cells, debris and medium/large vesicles. The resulting supernatant was applied to a 0.22 μm filter and ultracentrifuged at $110,000 \times g$ in a 70Ti fixed-angle rotor placed in a Beckman Coulter Optima LE-80K ultracentrifuge for 70 min at 4 °C to pellet EVs. The EV collections were washed once, resuspended in PBS and centrifuged a second time at $110,000 \times g$ for 16 h at 4 °C. The EV pellet was finally resuspended in DMEM or PBS.

Serum EVs (Serum-EVs) were isolated using an EV extraction kit for serum in accordance with the manufacturer's instructions (cat#4478360, Invitrogen). Briefly, serum was centrifuged at $2000 \times g$ for 30 min to remove cells and debris, the resulting supernatant was transferred to a new tube. Then 20 μl extraction reagent was added to a 100 μl serum sample and incubated at 2 °C to 8 °C for 30 min, followed by $10,000 \times g$ for 10 min to collect EV pellet. The resulting pellets were resuspended in PBS and applied to a 0.22 μm filter to remove large EVs, and centrifuged at $110,000 \times g$ in a 55Ti horizontal rotor placed in by a Beckman Coulter Optima LE-80K ultracentrifuge for 16 h to pellet EVs and resuspended in PBS.

For the EV identification assay, 100 mg RT-BAT or Cold-BAT was used for EV collection, and then the EVs were suspended with PBS at the same volume for testing. For tail vein injections, C57BL/6 J mice aged 7-8 weeks were used to collect BDEV, each mouse received 80 μg or 160 μg BDEVs for three consecutive days as indicated in each experiment. Other details were described in each section.

### Transmission electron microscopy (TEM)
The isolated EVs suspension was dropped onto the copper grid with carbon film for 3–5 min, and then use filter paper to absorb the excess

liquid. Dropped 2% phosphotungstic acid (cat# G1102, Servicebio) on the copper grid to stain for 1–2 min, use filter paper to absorb excess liquid. After the grids were air-dried, observed in a HT7800 (HITACHI, Tokyo, Japan) transmission electron microscope operated at 120 kV.

### Nanoparticle tracking analysis (NTA)
To analyse the size distribution of EVs, a dark-field microscope Nanosight NS300 (Malvern Panalytical, Amesbury, UK) was used. The samples were first diluted with saline to attain a concentration of $10^7$–$10^8$ particles/mL for analysis. Each sample was measured in triplicate at the camera and analyzed by NanoSight NTA3.2 software.

### EVs incubation assay
For fluorescent labelling experiments, BDEVs isolated from the cultured iBAT were labelled with Dil-C16 (red) per the manufacturer's instructions (cat# C7001, Invitrogen). Primary hepatocytes were isolated from WT mice as previously described and were seeded in 12-well plates. Before experimental manipulations, hepatocytes were rested for 4 h. A total of $3.5 \times 10^7$ BDEVs were labelled with the fluorescent dye Dil and incubated with $1.5 \times 10^5$ hepatocytes for EV tracking. After 6 hours of co-culture, the cells were washed three times with sterile PBS to remove non-internalized EVs. The hepatocytes were fixed with 4% paraformaldehyde for 20 min at room temperature. EV uptake by hepatocytes was captured by confocal laser scanning microscope LSM880 (Zeiss, Oberkochen, Germany) using 561 nm fluorescent channel. The merged images were created by overlaying DAPI and red fluorescent channels.

For functional gene and protein analysis, primary cultured hepatocytes were seeded at the number of $4 \times 10^6$ cells per 10 cm dish, and equal amounts of particles or proteins of BDEV and hepatocytes were co-incubation for 48 h, followed by functional analysis. Other details were described in each section. EV-free FBS was prepared and used in all EV incubation experiments.

### Transfection
Primary adipocytes were seeded into 6-well plates overnight and transfected the next day using INTERFERin (cat#409-10, Polyplus Transfection) according to the manufacturer's instructions. Mimic-miR-378a-3p and cy3- labelled miR-378a-3p were purchased from RiboBio (Guangzhou, China). To increase the expression of miR-378a-3p, 10 pmol of mimic-miR-378a-3p was transfected into $1 \times 10^6$ adipocytes. Equal scrambled mimic-miRNA was used as a negative control, 6 h later, the spent Opti-MEM was replaced with fresh DMEM/F12 culture medium. For confocal microscopy analysis of fluorescently labelled miRNAs, primary adipocytes were transfected with fluorescently labelled miR-378a-3p, the isolation and co-culture of BDEV with primary hepatocytes are as previously described. The hepatocytes were washed, fixed and then observed using a confocal microscope LSM 880 (Zeiss, Oberkochen, Germany). The excitation wavelengths

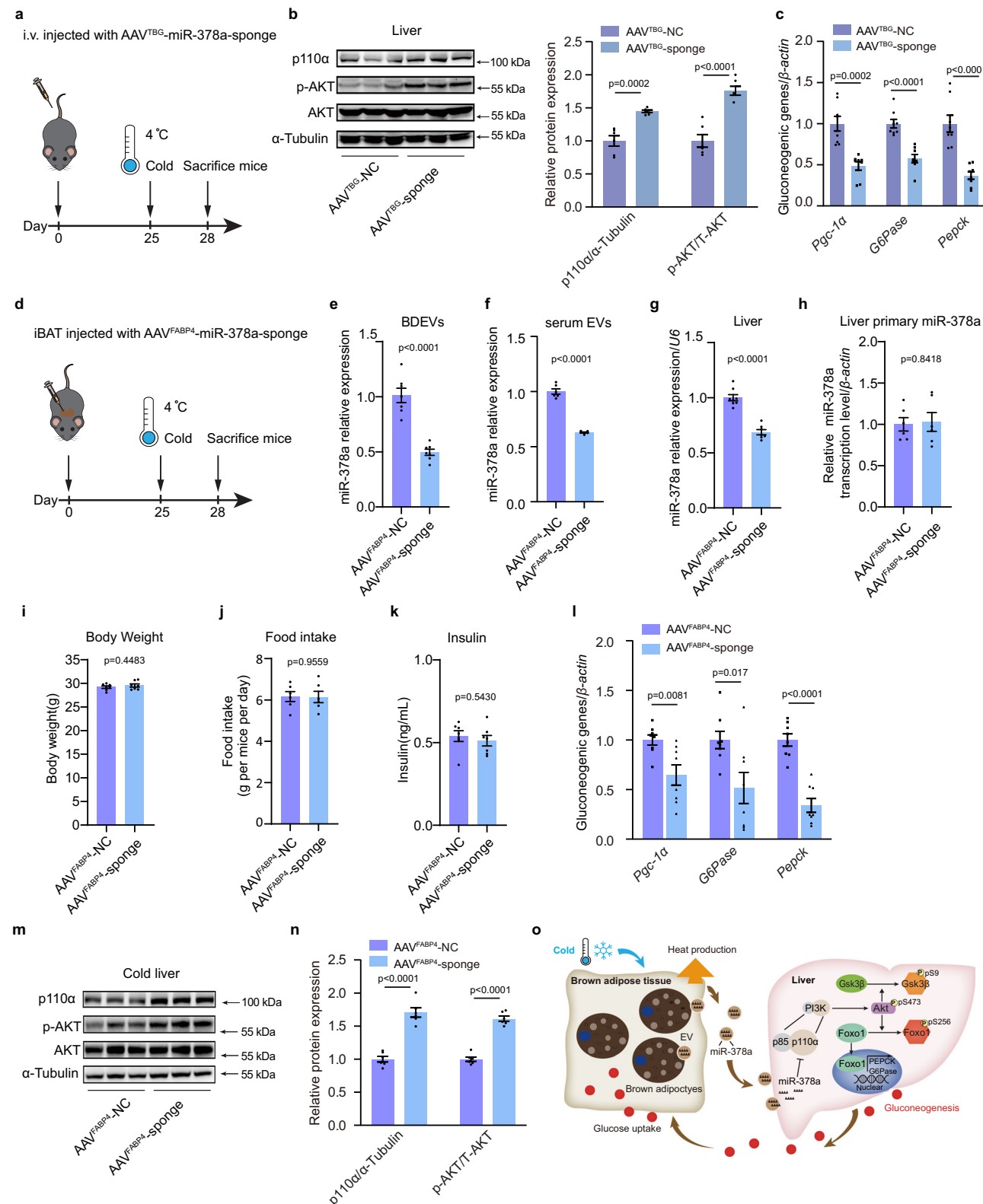

were 405 nm for DAPI and 532 nm for fluorescently labelled miR-378a-3p. For primary hepatocyte transfection with inhibitor, primary hepatocytes were isolated from WT mice as previously described and were seeded at the number of $4 \times 10^6$ cells per 10 cm dish. Before experimental manipulations, hepatocytes were rested for 4 hours. The Cold-BDEVs were isolated from the cultured iBAT from cold-exposed mice as described in the Methods section. The hepatocytes were transfected with different doses of anti-miR-378a (0, 10 nM, 25 nM,

50 nM, 100 nM, 200 nM) and co-incubation with 32 µg Cold-BDEVs for 48 h. All experiments were performed in triplicate wells for each condition and repeated 3 times independently.

## Cell glucose production

Collect RT- or Cold-BDEVs as described above, hepatocytes were first treated with BDEVs for 48 h. Then, the culture medium was replaced with HBSS supplemented with 100 nM insulin. Twenty minutes later,

**Fig. 8 | Depletion of miR-378a-3p in BAT impairs cold-induced hepatic gluconeogenesis. a–c** Liver-specific reduction of miR-378a-3p induced with AAV construct under the liver tissue-specific promoter TBG (AAV$^{TBG}$-sponge), a total of 100 μl of 1 × 10$^9$ Vg/μl of each AAV was injected through the tail vein for eight-week-old male C57BL6/J mice. Control mice were injected with AAV$^{TBG}$-NC. **a** Schematic figure. **b** Western blotting for p-AKT, AKT, and p110α in the liver (left panel) and densitometry analysis (right panel), α-Tubulin was used as an internal control ($n = 6$ each group, from 2 independent experiments). **c** qPCR analysis of the mRNA levels of gluconeogenic genes in the liver ($n = 8$ for each group, from 2 independent experiments). **d–n** iBAT-specific reduction of miR-378a-3p induced with adeno-associated virus constructed under the adipose tissue-specific promoter FABP4 (AAV$^{FABP4}$-sponge), a total of 50 μl of 1 × 10$^9$ Vg/μl of each AAV was injected into the iBAT in situ of eight-week-old male C57BL6/J mice. Control mice were injected with

AAV9- scrambled Control (NC). qPCR analysis of the relative miR-378a-3p expression level in **e** BDEVs ($n = 7,7$); **f** serum EVs ($n = 6,6$); and **g** livers ($n = 8,7$). **h** Relative level of primary miR-378a-3p in the liver ($n = 6$ for each group, from 2 independent experiments). **i** body weight ($n = 7,7$); **j** food intake ($n = 6,6$); **k** serum insulin level ($n = 7$ each group, from 2 independent experiments). **l** qPCR analysis of the mRNA levels of gluconeogenic genes in the liver ($n = 8$ each group, from 2 independent experiments). **m** Western blotting for p-AKT, AKT, p110α and α-Tubulin in the liver and **n** densitometry analysis of p-AKT, and p110α. α-Tubulin was used as an internal control ($n = 6$ for each group, from 2 independent experiments). **o** Graphic illustration of the cold-activated BDEVs-miR-378a-3p stimulates hepatic gluconeogenesis. Mice were fasted for 4 h before the test. Data presented as mean ± sem. All statistical analysis was performed using two-sided unpaired t-test. Source data are provided as a Source Data file.

the cell supernatant was collected for gluconeogenesis analysis using the Glucose Assay Kit (cat# F006, Jiancheng bioengineering). The assay was performed according to the manufacturer's instructions, and normalized by total cellular protein content.

## Body temperature

To measure mouse rectal body temperature, dip the temperature probe (BAT-12, Physitemp, New Jersey, USA) in Vaseline and place it in the rectum of each mouse. All measurements were performed at room temperature.

## Glucose uptake measurement

After being placed at 4 °C for 72 h and fasted for 12 h, the mice were intraperitoneally injected with 1 mg/mice 2-NBDG (cat#N-13195, Sigma), and blood and tissues were quickly collected at 60 min after injection. The determination and calculation method of glucose uptake measurement refers to Ye et al. previous work[50]. Cryostat sections (6-μm thick) of the iBAT from mice and stained with DAPI (1:1500) (cat#P36935, Invitrogen) and examined using a LSM 880 confocal microscope (Zeiss, Oberkochen, Germany).

## Tolerance tests

Pyruvate-tolerance test (PTT) were carried out on animals that had been fasted overnight for 16 h. Each animal received 1.5 g/ kg body weight of pyruvate via intraperitoneal (i.p.) injection after determination of fasted blood glucose levels. A glucometer (cat# 590, Yuwell, PRC) was utilized to determine the blood samples collected from the tail vein at 0,15,30,60 and 120 min posterior to injections. Mice were kept at ambient temperature throughout these assays.

## iBATectomy

As previously described[6], under isoflurane inhalation anesthesia, a small surgical incision was made in each mouse, carefully dissect BAT tissue. To prevent excessive bleeding after tissue resection, sterilized forceps were used to pinch the blood vessels in iBAT. The incision was then sutured with sterile surgical suture and the wound was disinfected. Sham operated mice were used as control group. All mice were recovered at ambient temperature for 10-day before subsequent experiments.

## Injection of GW4869 in vivo

For cold exposure of GW4869 treatment, 100 μl of GW4869 (3 mg/kg) (cat# D1692, Sigma) was either systematically (intraperitoneally) or locally injected into the BAT by 5-8 point-injections. After three days of injections, the mice were sacrificed 4 h after the last injection. For GW4869 treatment in mice fasted overnight (16 h fasting), 100 μl of GW4869 (3 mg/kg) was also either systematically (intraperitoneally) or locally injected into the BAT 4 h before sacrifice.

## PalmGFP lentiviruses

The recombinant viral plasmids encoding lentiviral particles and the packaging plasmids pGag/Pol, pRev and PVSV-G were constructed and prepared by GenePharma (Shanghai, China). As previously described[51], to generate fluorescent EV reporters, a palmitoylation signal (MLCCMRRTKQ) of growth cone-associated protein (GAP43) was genetically fused to the NH2 terminus of GFP (PalmGFP). PalmGFP sequences were inserted at XhoI and Bam HI sites of LV11 lentivector plasmid to obtain recombinant shuttle plasmids. 293 T cells were then stably transduced with a lentivirus vector encoding either PalmGFP (293T-PalmGFP) to examine EV reporter expression. To further examine whether PalmGFP labels EVs in association with membranes, we used PalmGFP to infect primary cultured brown adipocytes. For iBAT injection in situ, eight-week-old C57BL/6 J male mice were anesthetized with isoflurane, 100 μl of lentiviral particles (1×10$^9$ lentiviral transducing particles (TU)/ml) were administered directly by sight into the brown adipose tissues by 5-8 point-injections at the indicated time points and the virus-infected mice were housed at RT or cold-exposed for 72 h prior to sacrifice.

## Adeno-associated virus recombination and administration

All adeno-associated virus (AAV), including AAV9-FABP4-miR-378a-ZsGreen (AAV$^{FABP4}$-miR-378a), AAV9-FABP4-miR-378a sponge-ZsGreen (AAV$^{FABP4}$-miR-378a-sponge), and AAV8-TBG-miR-378a sponge-ZsGreen (AAV$^{TBG}$-miR-378a-sponge) were constructed, amplified, and purified by HANBIO Biotechnology Co. Ltd (Shanghai, China). Briefly, miR-378a-3p (AGGGCTCCTGACTCCAGGTCCTGTGTGTTACCTCGAAA TAGCACTGGACTTGGAGTCAGAAGGCCT) was cloned into the pHBAAV-FABP4-ZsGreen vector, followed by sequencing confirmation, then the shuttle vector was packaged into capsids from AAV serotype 9 (AAV9-FABP4-miR-378a-ZsGreen, used in BAT); miR-378a-sponge (CCTTCTGACTGGAGTCCAGTtatacCCTTCTGACTGGAGTCCA GTacatcCCTTCTGACTGGAGTCCAGTtcttcaCCTTCTGACTGGAGTCCA GT) was cloned into the pHBAAV-FABP4-ZsGreen and pHBAAV-TBG-MCS-P2A-ZsGreen vectors, respectively. After sequencing confirmation, the shuttle vector pHBAAV-FABP4-378a sponge-ZsGreen was packaged into capsids form AAV serotype 9 (AAV9-FABP4-miR-378a sponge-ZsGreen, used in BAT), while the shuttle vector pHBAAV-TBG-MCS-P2A-ZsGreen was packaged into capsids from serotype 8 (AAV8-TBG-miR-378a sponge-ZsGreen, used in liver). Viral particles were purified and then titered by qRT-PCR analysis. For adipose-specific knockdown and overexpression of miR-378a-3p, 50 μl of 1 × 10$^9$ Vg/μl of AAV$^{FABP4}$-miR-378a or AAV$^{FABP4}$-miR-378a -sponge was injected into the iBAT of mice. For liver-specific knockdown of miR-378a-3p, a total of 100 μl of 1 × 10$^9$ Vg/μl of AAV$^{TBG}$-miR-378a-sponge was injected through the tail vein for each mouse. The corresponding control mice were injected with the same amount of NC vector. As the schematic diagram indicated in each experiment, 28 days post AAV injection, mice were sacrificed for tissue collection and biochemical study.

## Metabolic index

Mouse Insulin ELISA (cat#10-1247-01, Mercodia) and Glucagon ELISA (cat#10-1281-01, Mercodia) were employed to identify serum insulin and glucagon contents, respectively. All operations were performed in accordance with the specification and eventually identified on a spectrophotometer (Thermo Fisher Scientific, America).

## Histology

iBAT were fixed in 4% paraformaldehyde (PFA) for 24 h, dehydrated, transparentized and embedded in paraffin. The paraffin-embedded tissue sections of 5 µm thickness were dyed with haematoxylin and eosin (H&E) (cat#DH0006, Leagene Biotechnology) for morphological observation. Sections of H&E staining were captured under a virtual slide microscope (VS120, Olympus). All quantifications were performed using Image-Pro plus software 6.0 (Media Cybernetics), analyses of which were performed by two or more researchers independently.

## Immunofluorescence (IF) staining

The lentivirus-infected tissues were soaked in 4% PFA for 24 h, cryoprotected in 15% sucrose for 1 h, dehydrated in 30% sucrose for 12 to 72 h and then embedded in OCT. Cryostat sections (6-µm thick) of the iBAT and liver were washed in PBS for three times 5 min, then followed by 10 min in PBST (0.5% Triton X-100) before blocking in 5% bovine serum albumin in PBST (0.5% Triton X-100) for 1 h. Liver sections were incubated with anti-CK18 (cat#A19778,1:100,ABclonal) and iBAT sections were incubated with anti-UCP1 (cat#72298,1:100,CST) antibodies overnight at 4 °C, then incubated with Donkey anti-Rabbit IgG (H + L) Highly Cross-Adsorbed Secondary Antibody, Alexa Fluor™ 594 (cat# A-21207, 1:1000, Invitrogen) for 1 h (room temperature). The co-immunostaining sections and the other peripheral tissues sections (10-µm thick) and brain sections (25-µm thick) were washed in PBS for three times 5 min and stained with DAPI (1:1500) (cat# P36935, Invitrogen). The sections were examined using a LSM 880 confocal microscope (Zeiss, Oberkochen, Germany). ZEISS ZEN 3.6 software was used for image processing, 5 randomly selected visual fields/section of at least 10 nonsequential sections per mouse/time point were analysed.

## RNA isolation and quantitative RT−PCR detection

Total RNA from cultured cells or tissues was extracted using RNAiso Plus (cat#9109, Takara) according to the manufacturer's instructions. 1 µg RNA was used to synthesize the first-strand cDNA using the HiScript III RT SuperMix (cat#R323, Vazyme). The resultant cDNA was analyzed by quantitative real-time PCR performed with a Light-Cycler®480 (Roche Diagnostics, Mannheim, Germany) using ChamQ Universal SYBR qPCR Master Mix (cat# Q711, Vazyme). Thermocycle conditions are 95 °C for 5 min, then followed by 40 cycles of 95 °C for 15 s, 60 °C for 30 s, and 72 °C for 30 s. The relative mRNA expression was determined after normalization to $\beta$-actin using the $2^{-\Delta\Delta CT}$ method. Primer sequences are shown in Supplementary Table 2.

## miRNA expression analysis

For the microarray analysis, independent pooled subcutaneous fat (scWAT), visceral fat (viscWAT) and iBAT tissue samples were analysed from 8-week-old male C57BL/6 J mice. Each sample comprised a pool of iBAT from 5 animals. Total RNA from each pooled sample was isolated using the TRIzol method for Affymetrix miRNA microarray analysis (CapitalBio Corp., Beijing, China). Procedures were performed as described on the website of CapitalBio (http://www.capitalbio.com). Briefly, 50−100 µg of total RNA was used to extract miRNA with a miRNA Isolation Kit (Ambion Inc. Texas, USA). Biotin-labelled miRNAs were used for hybridization on each miRNA microarray chip containing probes in triplicate. Raw data were normalized to U6 and analysed using GenePix Pro 4.0 software (Axon Instruments, PA, USA). The following criteria were used to screen the miRNAs from the array data set: miRNAs with signal intensity greater than 30 were selected to avoid weak signal data. The miRNAs co-expressed in the three adipose tissues were screened, and the results were presented in the form of Venn. Mice iBAT microarray data in cold exposure group and control group were obtained from GEO database (GSE 41306)[35]. MiRNAs from the cold exposure groups were each compared with those from the control group; after normalization, miRNAs that showed opposite expression ratios in the cold exposure groups were selected. The data were presented as a heat map with colour indicating the fold-change for each miRNA. Q-PCR validation was performed on the miRNAs screened above. TaqMan miRNA probes (Applied Biosystems, CA, USA) were used to quantify the reported miRNA expressions levels, including miR-378a-3p, the miR-193b-3p cluster, miR-192, miR-21, miR-223 and U6 snRNA.

## Western blot analysis

For insulin signalling analysis, the livers were removed and frozen in liquid nitrogen and proteins were extracted for western blotting analysis. Cells or tissues were lysed in ice-cold RIPA buffer (cat#P0013B, Beyotime) supplemented with protease and phosphatase inhibitor cocktail (cat#P1048, Beyotime). Cell membrane proteins were extracted according to the manufacturer's instructions (cat#P0033, Beyotime). Protein concentration was measured by the Pierce™ BCA protein assay kit (cat#23227, Thermo Scientific). The anti-UCP1 antibody (cat#72298, 1:1000), anti-α-Tubulin antibody (cat#3873,1:1000), anti-p-PKM2 antibody (cat#3827,1:1000), anti-PKM2 antibody (cat#4053,1:1000), anti-p-AKT antibody (cat#4060,1:1000), anti-AKT antibody (cat#9272,1:1000), anti-p-GSK3β antibody (cat#9323,1:1000), anti-GSK3β antibody (cat#12456,1:1000), anti-p-FOXO1 antibody (cat#84192,1:1000), anti-FOXO1 antibody (cat#2880,1:1000), anti-p110α antibody (cat#4249,1:1000), anti-p-IR antibody (cat#3024,1:1000), anti-IR antibody (cat#3020,1:1000), anti-Alix antibody (cat#2171,1:1000) and anti-CD9 antibody (cat#98327,1:1000) were purchased from Cell Signalling Technology. The anti-GLUT4 antibody (cat#07-1404,1:1000) was purchased from Millipore. The anti-PGC-1β antibody (cat#sc-373771,1:1000) and anti-CD63 antibody (cat#sc5275,1:1000) were purchased from Santa Cruz Biotechnology. The anti-CD81 antibody (cat#ab109201,1:1000), anti-DICER antibody (cat#ab14601,1:1000), and anti-ALBUMIN antibody (cat#ab207327,1:2000) were purchased from Abcam. The anti-LAMIN A/C antibody (cat# A19524,1:10000) was purchased from ABclonal. Results were analyzed using ImageJ. The signal of each image was normalized such that the average of all control samples equaled 1.0. Other details of antibodies are described in Supplementary Table 3.

## Statistics

GraphPad Prism 9 software were used for statistical analysis, firstly, normal distribution of data was assessed by either D'Agostino-Pearson's normality test ($n > 7$) or Shapiro-Wilk's test ($n \leq 7$). Upon confirmation of normal distribution, unpaired student's t-test (two-tailed), one-way ANOVA analysis followed by Bonferroni's multiple comparisons test, or two-way ANOVA analysis followed by Bonferroni's multiple comparisons test was performed. Where normal distribution failed, a Kruskal-Wallis one-way ANOVA analysis followed by Dunn's multiple comparisons test or a Mann-Whitney U test (two-tailed) was performed. All values are presented as the mean ± sem. All data with significant difference were analyzed by GPower3.1 to obtain effect size and power value. Tests used for statistical analyses are shown in figure legends and statistical details can be found in Source Data.

## Reporting summary

Further information on research design is available in the Nature Portfolio Reporting Summary linked to this article.

## Data availability

The miRNA microarray data generated by Trajkovski et al. (GSE 41306)[35] and by us (GSE 217222) are deposited at the Gene Expression Omnibus repository. Source data are provided in this paper.

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

## Acknowledgements

This work was supported by grants from the National Natural Science Foundation of China (31972912, 32371245), the Natural Science Foundation of Jiangsu Province (BK20211153), the Fundamental Research Funds for the Central Universities (020814380173), the CAMS Innovation Fund for Medical Sciences (No. CIFMS-2021-I2M-5-015).

## Author contributions

X.J., C.Y.Z., and K.Z. designed the experiments and oversaw all aspects of study conduct and manuscript preparation. J.X., L.C., J.W., S.Z., H.Z., S.K., X.C., Y.S. performed the experiments. J.X., L.C., J.W. J.L., A.V., C.Y.Z., L.L. analyzed the data. L.L., X.J., K.Z., and C.Y.Z. contributed reagents/materials/analysis tools. X.J., J.X., and A.V. wrote and edited the paper.

## Competing interests

The authors declare no competing interests.
