## [Peer Review File · Nature Communications]

Cold-activated brown fat-derived extracellular vesicle-miR-378a-3p stimulates hepatic gluconeogenesis in male miceREVIEWER COMMENTS

Reviewer #1 (Remarks to the Author):

Cold-activated brown fat-derived extracellular vesicle-miR-378a-3p stimulates hepatic gluconeogenesis to maintain glucose homeostasis.

In this work authors show how activated brown adipose tissue (BAT) can reprogram systemic glucose metabolism by promoting hepatic gluconeogenesis during cold stress through the secretion of extracellular vesicles. More precisely, they affirm that cold exposure facilitates the selective packaging of a BAT enriched miRNA (miR-378a-3p) into extracellular vesicles and its delivery into the liver. They describe that once in the liver, BAT-shed miR-378a-3p enhances gluconeogenesis by targeting the subunit of PI3K, p110 α . Moreover, they show how miR-378KO mice have reduced hepatic gluconeogenesis during cold exposure, while restoration of miR-378a-3p in iBAT induces the expression of gluconeogenic genes in the liver. They conclude that they have provided evidences to suggest that miRNAs transported in brown adipose-shed vesicles as stress-induced batokine able to coordinate systemic glucose homeostasis. Thus, in this article, they propose the miR-378a-3p as a key actor on interorgan communication as novel endocrine function of BAT preventing hypoglycemia during cold stress.

This work is of great interest, with noteworthy results and of significance to the field since there are many evidences suggesting EVs as main key players in energy homeostasis regulation; however, there is still little knowledge about the molecular events implicated in EV signaling in metabolism, and there is a great need to develop their role in this field. Thus, in addition to being innovative, this work is experimentally very elegant and well designed, well planned, and implemented. The work supports the conclusions and claims with no flaws in the data analysis, interpretation, and conclusions. The methodology is sound and appropriated following the field standards; moreover, methods are explained in detail for the work to be reproduced.

This reviewer has just few minor comments:

Methods:

-Authors do not include sex and gender considerations for studies involving vertebrate animals. The methods section should include whether sex and/or gender were considered in the study design and whether sex and/or gender of participants was determined based on self-report or assigned (and methodology used).

-The term Extracellular vehicle (EV) has to be revised; it is strongly recommended to follow the international guidelines for the study and nomenclature of extracellular vesicles. Please note that EV standardly stands for Extracellular Vesicles, not Vehicles. In this regard, authors should state and make sure that they have followed the minimal information for studies of extracellular vesicles as stated by the International Society for Extracellular Vesicles (Théry et al., J Extracel Vesicles 2018).

-In the results, line 147 authors affirm they isolate EVs via ultrafiltration; however, in the method section, they describe they were isolated via differential centrifugation.

-Why do the authors choose 24h incubation period for EVs collection?

-For EVs incubation assays: authors should explain how they calculated the BAT-EVs doses (concentration) to treat hepatocytes; moreover, it is necessary to explain the elected 6h incubation time.

-Treatment of mice with 80ug of BDEV has to be explained; please note that equal amounts of vesicles may have different amount of cargo proteins.

Results:

Figure 1: please indicate tissue type for each analysis/graph indicating BAT or liver to make interpretation of the results easier for the reader.

Figure 3: 3A. Why did the authors incubate hepatocytes for 48 hours and 6 hours in Figure 2G? How the authors checked if BAT-EVs reach other peripheral tissues or CNS? Does the cold-BDEV injection in Figure 3E may have a role on white adipose tissue in these animals? Have the authors consider to check this same mechanism in animals under high fat diet?

Reviewer #2 (Remarks to the Author):

Brown adipose tissue (BAT) takes up a lot of circulating glucose to fuel thermogenesis under cold exposure. However, little is known about the underlying mechanism. In this manuscript, the authors showed that cold stress stimulated BAT to produce more extracellular vesicles (BDEVs), selectively packaged more miR-378a-3p, a known metabolism-relevant miRNA encoded in the *Ppargc1b* gene, into the EV, and delivered to the liver to promote gluconeogenesis. The authors confirmed miR-378a-3p enhances gluconeogenesis by targeting p110 α , a subunit of PI3K, which has been well-studied in a 2014 paper (ncomms6684). Further, the authors confirmed that miR-378 KO mice display reduced hepatic gluconeogenesis during cold exposure, while restoration of miR-378a-3p in iBAT induces the hepatic expression of gluconeogenic genes. These findings did provide some novel insight into metabolic interorgan coordination. However, there are still some major concerns that need to be addressed.

1. It is unclear if the other components besides miR-378a-3p also contribute to elevated hepatic gluconeogenesis. How much does miR-378a contribute to the hepatic gluconeogenesis effect of Cold-BDEV? The primary hepatocytes should be treated with Cold-BDEV and different concentrations of miR-378a-3p inhibitor to check how the hepatic gluconeogenesis will be affected.
2. What will be the possible reason leading to the distinct responses of miR-378a-3p transcription in BAT and the liver to cold treatment?
3. If the EV inhibitor GW4849 is systematically or BAT locally injected, how will the hepatic miR-378a-3p level change after the cold treatment or fasting?
4. The AAV to specifically binding miR-378a-3p in BAT, such as AAV-FABP4-miR-378a-sponge, should be constructed, produced, and injected into the BAT and then check if the hepatic gluconeogenesis will be regulated or not.
5. Fig 4K, it should be critical to know the absolute level of miR-378a-3p (such as the counts per μ g of mass) included in the BDEV which was injected in the mice and the absolute level of miR-378a in the liver before and after the EV injection.
6. Line 252, "p110 α , the fundamental subunit of phosphoinositide 3-kinase (PI3K) that controls glucose metabolism was predicted to be a direct target of miR-378a-3p." However, in Ref 34 (<https://www.nature.com/articles/ncomms6684>), their title has clearly shown that "miR-378 targets p110 α and controls glucose and lipid homeostasis". p110 α is a known

target of miR-378 that doesn't need to be predicted.

Minor:

1. Fig 2I, it is unclear if other components besides EV will be transferred to the liver after the transduction of FABP4-Palm-EGFP lentivirus in BAT. QPCR should be performed to confirm whether GFP mRNA is detectable in the liver after the transduction.
2. Fig 2J-K, a BAT or liver-specific antibody should be used for co-immunostaining.
3. Fig 2H, the palmitoylated residue should be cysteine but no other amino acids, as in <https://www.nature.com/articles/ncomms8029> Fig1a.
4. Fig 4H-I, it is unclear if the level or activity of the miRNA biogenesis proteins such as Dicer will be elevated by cold in the mouse liver, leading to the increase of mature miRNAs level.
5. The reference for GSE41306 should be cited.
6. Line 548, AAV is not adenovirus. How these AAV vectors were constructed should be demonstrated.

Point by Point Response to Reviewers' Comments

Reviewer #1 (Remarks to the Author):

Cold-activated brown fat-derived extracellular vesicle-miR-378a-3p stimulates hepatic gluconeogenesis to maintain glucose homeostasis.

In this work authors show how activated brown adipose tissue (BAT) can reprogram systemic glucose metabolism by promoting hepatic gluconeogenesis during cold stress through the secretion of extracellular vesicles. More precisely, they affirm that cold exposure facilitates the selective packaging of a BAT enriched miRNA (miR-378a-3p) into extracellular vesicles and its delivery into the liver. They describe that once in the liver, BAT-shed miR-378a-3p enhances gluconeogenesis by targeting the subunit of PI3K, p110 α . Moreover, they show how miR-378KO mice have reduced hepatic gluconeogenesis during cold exposure, while restoration of miR-378a-3p in iBAT induces the expression of gluconeogenic genes in the liver. They conclude that they have provided evidences to suggest that miRNAs transported in brown adipose-shed vesicles as stress-induced batokine able to coordinate systemic glucose homeostasis. Thus, in this article, they propose the miR-378a-3p as a key actor on interorgan communication as novel endocrine function of BAT preventing hypoglycemia during cold stress.

This work is of great interest, with noteworthy results and of significance to the field since there are many evidences suggesting EVs as main key players in energy homeostasis regulation; however, there is still little knowledge about the molecular events implicated in EV signaling in metabolism, and there is a great need to develop their role in this field. Thus, in addition to being innovative, this work is experimentally very elegant and well designed, well planned, and implemented. The work supports the conclusions and claims with no flaws in the data analysis, interpretation, and conclusions. The methodology is sound and appropriated following the field standards; moreover, methods are explained in detail for the work to be reproduced.

This reviewer has just few minor comments:

Methods:

1. -Authors do not include sex and gender considerations for studies involving vertebrate animals. The methods section should include whether sex and/or gender were considered in the study design and whether sex and/or gender of participants was determined based on self-report or assigned (and methodology used).

Answer: We appreciate the reviewer for pointing out this critical issue and apologize for not clearly indicating the sex and gender considerations. We completely agree with the reviewer that we should clarify that only male mice were used in the present study. In the study design, male mice were selected primarily based on the following considerations: 1) gender differences in glucose

metabolism have been reported in mice due to the different expression pattern of sex hormones (Varlamov O et al., *Front Endocrinol.* 2015; Jan 19; 5:241. PMID 25646091). In particular, estrogen, which reaches a maximal (~60 pg/ml) in the proestrus stage and is reduced to ~20 pg/ml at estrus in female mice (Saito T et al., *Circ Res.* 2009 Aug 14; 105(4):343-352. PMID: 19608983), was shown to suppress gluconeogenesis (Yan H et al., *Diabetes.* 2019 Feb; 68(2):291-304. PMID: 30487265). Therefore, to avoid the possible influences caused by the estrogen cycle on glucose metabolism in female mice, male mice were chosen for the current study. Moreover, whether gender differences exist in response to cold exposure in mice is still controversial. According to a recent study on young adults, short-term cold exposure showed more pronounced hormonal changes in women than in men (Mengel L et al., *J Clin Endocrinol Metab.* 2020; May 105 (5): e1938-e1948. PMID:32144431). Therefore, to reduce the variations that may result from hormonal changes in female mice, only male mice were chosen for the current study. Our appreciation for the reviewer's emphasis on this critical issue. We have added considerations regarding to gender differences in the study design and indicated in the title and abstract that only male mice were used in the present study (line 2 in title, line 37 in abstract, line 105 in introduction, line 486-489 in Methods, highlighted).

2. -The term *Extracellular vehicle (EV)* has to be revised; it is strongly recommended to follow the international guidelines for the study and nomenclature of extracellular vesicles. Please note that EV standardly stands for *Extracellular Vesicles*, not *Vehicles*. In this regard, authors should state and make sure that they have followed the minimal information for studies of extracellular vesicles as stated by the International Society for Extracellular Vesicles (Théry et al., *J Extracel Vesicles* 2018).

Answer: Thank you very much for pointing out this mistake in the manuscript. Our sincere apology for mistakenly writing Extracellular Vesicles (EVs) as vehicle in the Methods section, we have corrected it in the revision (line 527 in Methods, highlighted). According to the reviewer's constructive suggestion, by following the information for studies of EVs as stated by the International Society of Extracellular Vesicles (Théry et al., *J Extracel Vesicle.* 2018), we characterized the physical properties of the EVs in the revision, detailed experimental procedure and characterization of EVs are as follows:

The EVs from the cultured interscapular brown adipose tissue (iBAT) were isolated as follows: the iBAT of mice was excised and transferred into 2 ml DMEM with 1% P/S and gently cut into 2 mm³ pieces. The chopped tissues were centrifuged at 1000 × g for 5 min at room temperature, pellets were resuspended in DMEM and transferred into a petri dish. After 30 min of incubation in DMEM containing 2% EV-free FBS, 0.5 mg/ml DNase I, 0.2 mg/ml RNase A and 1% P/S, the culture medium was exchanged. The tissues were incubated for 24 h at 37 °C, and 5% CO₂, and then the culture supernatant was collected for EV isolation by ultracentrifugation method. In brief, the medium was centrifuged at 300 × g for 10 min, 3000 × g for 20 min, and 10,000 × g for 30 min to remove tissues, cells, debris and large vesicles. The resulting supernatant was applied to a 0.22 μm filter and ultracentrifuged at 110,000 × g in a 70Ti fixed-angle rotor placed in a Beckman Coulter Optima LE-80K ultracentrifuge for 70 min at 4 °C to pellet EVs. The EV collections were washed once, resuspended in PBS and centrifuged a second time at 110,000 × g for 16 hours at 4 °C. The EV pellet was finally resuspended in DMEM or PBS (line 528-541 in Methods, highlighted). The physical properties of the EVs derived from iBAT (BDEVs) were characterized as follows:

Nanoparticle tracking analysis (NTA) revealed that the BDEVs showed the enrichment of small EVs, which were < 200 nm and peaked at ~ 107 nm and ~ 102 nm of RT-BDEVs and Cold-BDEVs (see Figure 2C below). Transmission electron microscopy (TEM) confirmed that the purified EVs exhibited a characteristic vesicular morphology, without significant differences in particle shapes between RT-BDEVs and Cold-BDEVs (see Figure 2B below). According to immunoblotting analysis, EV-associated proteins, including CD63, CD9 and ALIX were highly enriched in BDEVs, while LAMIN A/C and UCP1, proteins found abundantly in the nucleus and the mitochondrial inner membrane of brown adipocytes, were barely detected in BDEVs, consistent with the known characteristics of EVs (see Supplementary Figure S2E below).

Serum EVs (Serum-EVs) were isolated using an EV extraction kit for serum in accordance with the manufacturer's instructions (Invitrogen, cat#4478360). Briefly, serum was centrifuged at 2000 × g for 30 min to remove cells and debris, and the resulting supernatant was transferred to a new tube. Then, 20 µl of extraction reagent was added to a 100 µl serum sample and incubated at 2 °C to 8 °C for 30 minutes, followed by 10,000 × g for 10 min to collect the EV pellet. The resulting pellets were resuspended in PBS, applied to a 0.22 µm filter to remove large EVs, centrifuged at 110,000 × g in a 55Ti horizontal rotor placed in a Beckman Coulter Optima LE-80K ultracentrifuge for 16 h to pellet EVs and resuspended in PBS (line 542-549 in Methods, highlighted). The physical properties of the EVs isolated from serum (Serum-EVs) were characterized as follows: TEM and NTA revealed typical vesicular characteristics of EVs (see Supplementary Figure S6A-S6B below). Protein markers of EVs, including CD63, CD 9 and ALIX were also highly enriched in serum-EVs, while ALBUMIN, which is abundantly expressed in serum, was absent in the serum-EVs, confirming the purity of the serum-EVs (see Supplementary Figure S6C below).

3. -In the results, line 147 authors affirm they isolate EVs via ultrafiltration; however, in the method section, they describe they were isolated via differential centrifugation.

Answer: We appreciate the reviewer for pointing out this inconsistency in the manuscript. We are very sorry for having wrongly described ultracentrifugation as ultrafiltration or differential centrifugation, and we have corrected the mistakes in the revised manuscript (line 158 in results and line 528 in Methods, highlighted). As described in the Methods section, the cultured medium was centrifuged at $300 \times g$ for 10 min, $3000 \times g$ for 20 min, and $10,000 \times g$ for 30 min. The resulting supernatant was applied to a $0.22 \mu\text{m}$ filter (Millipore) and ultracentrifuged at $110,000 \times g$ for 70 min at 4°C to pellet EVs. The EV collections were washed once, resuspended in PBS and centrifuged a second time at $110,000 \times g$ for 16 hours at 4°C . The EV pellet was finally resuspended in DMEM or PBS (line 528-541 in Methods, highlighted). Therefore, the BDEVs in the supernatant of the cultured iBAT was isolated by the ultracentrifugation method.

4. -Why do the authors choose 24h incubation period for EVs collection?

Answer: Thanks the reviewer for raising up this issue about the selection of the timepoint of EV collection. Since the presence of active BAT was characterized in adult humans in 2009, a significant increase in research interest has focused on the metabolic regulatory role in BAT, including several studies that working on BAT-derived EVs; however, a standardized methodological approach, especially regarding BAT-derived EVs has not yet been fully established. Therefore, according to the literature research and our preliminary experimental data, a 24 h incubation period was selected for BDEVs collection in the current study as follows:

Based on previously published articles on BAT-derived EVs, a 2h (Chen T et al, *Nat Commun.* 2016. Apr 27; 7:11420. PMID:27117818) or 24 h (Rosina M et al, *Cell Metab.* 2022 Apr5;34(4):533-548.e12. PMID:35305295; Zhou X et al, *Theranostics.* 2020. Jul 9;10 (18):8197-8210. PMID:32724466) incubation period was selected for EV collection. Therefore, in our study, a time-course experiment was conducted to measure the EV concentration collected at different time points.

After 2 h, 6 h, 12 h, 24 h and 48 h of incubation, BDEVs were collected from the culture medium of BAT. Notably, the BDEV concentration (indicated by the EV particle amounts) gradually increased within 24 hours and reached a plateau in both RT-BAT and Cold-BAT group (see Supplementary Figure S2B below, line 159-162, highlighted). Our experimental purpose was to compare the differences in BDEVs from RT- or cold-exposed mice, including the quantification of BAT-secretion ability and the function of BDEVs from each group. According to the experimental data, a short collection period may not yield enough EV samples, whereas a longer incubation at 37 °C may lead to significant differences from physiological conditions. In addition, during the 3-day cold exposure period, the BDEVs were considered to be continuously released. Therefore, to mimic physiological conditions, a 24 h incubation period was chosen to collect BDEVs either from RT- or cold-exposed BAT, and each mouse received BDEVs three times (24 h each i.v. injection) to study the possible function of the Cold-BDEVs.

5-For EVs incubation assays: authors should explain how they calculated the BAT-EVs doses (concentration) to treat hepatocytes; moreover, it is necessary to explain the elected 6h incubation time.

Answer: Thanks the reviewer for pointing out these important issues. We apologize for not clearly clarifying the concentrations and incubation times of BDEVs used in different experiments. Generally, there are two parts of experiments referring to BDEV incubation assays. In Figure 2G and Supplementary Figure S2F, the experimental purpose was to track whether the EVs derived from BAT could be efficiently taken up by primary cultured hepatocytes; while in Figure 3A-D, the BDEVs were co-incubated with hepatocytes to access the influences of BDEVs on hepatocytes gluconeogenesis. Therefore, based on the purpose of each experiment, different incubation times and BDEV doses were used to treat the hepatocytes. The experimental details and results are as follows:

Considering that the selection of dosage treatment is a key issue to mimic physiological conditions in the *in vitro* experiments, the iBAT weight was measured in either control or cold-exposed mice. The iBAT weight was ~100 mg in both control and cold-exposed mice, indicating that although 72 h of cold exposure robustly activated the thermogenic function of iBAT, the weight of the iBAT mass was not markedly changed yet (Supplementary Figure S2A; line 157-158, highlighted). Therefore, 100 mg of isolated iBAT (approximately the amount of iBAT from each mouse) from

control or cold-exposed mice was cultured for BDEV collection. After 24 h of incubation, BDEVs were isolated from the cultured RT-BAT or Cold-BAT by ultracentrifugation. As revealed by NTA, we found that Cold-BAT released significantly more EV particles than RT-BAT (see Figure 2D below, 1×10^{10} Cold-BDEV vs. 4×10^9 RT-BDEVs). To mimic the physiological conditions, the primary hepatocytes isolated from each mouse liver were treated with BDEVs secreted from cultured RT-BAT or Cold-BAT (100 mg). Therefore, primary hepatocytes were also isolated from mice by collagenase perfusion as described in the Methods section (line 517-525). Approximately 2×10^7 cells hepatocytes were isolated from each mouse liver, and the cells were seeded 4×10^6 cells per 10cm dish (~1/5 hepatocytes isolated from each mouse liver) and 1.5×10^5 cells per well for 12-well plates, respectively.

As the purpose of the EV incubation experiments was to track whether BDEVs could be effectively taken up by recipient hepatocytes, based on the above calculation, 3.5×10^7 RT-BDEVs were labeled with the fluorescent dye Dil and incubated with 1.5×10^5 hepatocytes for EV tracking. According to a literature report and our previous experimental data, the fluorescent signal of the labeled-EVs could be detected in the recipient cells as early as 2 h post coincubation, and most of the studies choose a 6-10 h incubation period for the EV tracking assay (Saha B et al, *Hepatology*. 2018 May; 67(5):1086-2000. PMID: 29251792; Zhang Y et al, *Mol Cell*. 2010 Jul 9; 39(1):133-44. PMID: 20603081; Lin L et al, *Nat Metab*. 2023 May; 5(5): 821-841. PMID: 37188819). Thus, we performed a time-course study to track the fluorescent signals in the hepatocytes. Briefly, the labeled BDEVs were added to the culture medium of hepatocytes, and after 2 h, 4 h, 6 h, and 12 h of incubation, the cells were observed by laser scanning confocal microscopy. As indicated by the red fluorescence staining of the hepatocytes, a low-level fluorescence signal could be detected in hepatocytes after 2 h of incubation. With increasing incubation time, the fluorescent signal gradually increased and reached the plateau phase after 6 h of incubation (see Supplementary Figure S2F below; line 178-181, highlighted). Therefore, 6 h incubation was chosen for BDEV tracking in the revision (line xx-xx, high).

Supplemental Figure S2F Confocal microscopy image of primary cultured hepatocytes coincubation with fluorescent Dil-labelled BDEVs for indicated time points. Briefly, 3.5×10^7 BDEVs were labeled with the fluorescent dye Dil and incubate with 1.5×10^5 hepatocytes for EV tracking. After 2 h, 4 h, 6 h, and 12 h of co-culture, the cells were washed three times with sterile PBS to remove non-internalized EVs. Hepatocytes were washed, fixed, and observed under confocal microscopy. Scale bar: 20 μm .

While the purpose of incubating BDEVs with hepatocytes in Figure 3A-D and Supplementary Figure S4A-S4C was to compare the effects of different BDEVs on hepatocyte functions, so the BDEV doses used to treat the hepatocytes were as follows:

1) As we mentioned above, cold exposure significantly increased BDEV secretion. For 24 h incubation, 4×10^9 RT-BDEVs particles ($\sim 80 \mu\text{g}$ RT-BDEV protein) and 1×10^{10} particles Cold-BDEVs ($\sim 160 \mu\text{g}$ Cold-BDEV protein) were isolated from the culture medium from equal amounts (100 mg) of RT-BAT or Cold-BAT, and the primary cultured hepatocytes were co-incubated with BDEVs for 48 h for functional gene and protein analysis. Thus, to mimic the physiological conditions, 1.6×10^9 RT-BDEVs ($\sim 1/5$ of the total EVs isolated from cultured control BAT $\times 2$ days, $32 \mu\text{g}$ RT-BDEVs) and 4×10^9 Cold-BDEVs ($\sim 1/5$ of the total EVs isolated from cultured cold activated BAT $\times 2$ days, $64 \mu\text{g}$ Cold-BDEVs) were administered to 4×10^6 primary cultured hepatocytes ($\sim 1/5$ hepatocytes isolated from each liver). The BDEVs and hepatocytes were coincubation for 48 h, followed by functional analysis. Compared with the cells treated with 1.6×10^9 RT-BDEVs ($32 \mu\text{g}$ RT-BDEVs), the phosphorylation level of AKT was dramatically decreased in the hepatocytes administered 4×10^9 Cold-BDEV ($64 \mu\text{g}$ Cold-BDEVs), concurrent with markedly increased gluconeogenic genes (see Supplementary Figure S4A-S4C below; line 219-227, highlighted).

2) Since the above observations may result from the increased BDEV particle numbers or the elevated total protein amounts, we still wondered whether treatment with equal amounts of Cold-BDEV particles or proteins would also have similar effects. Therefore, 1.6×10^9 Cold-BDEVs (equal particle number as RT-BDEVs) were also applied to 4×10^6 hepatocytes and incubated for 48 h. Although the reduction was not as dramatic as those administered with $\sim 4 \times 10^9$ cold-BDEVs, markedly decreased levels of p-AKT were also observed in cells treated with $\sim 1.6 \times 10^8$ cold-

BDEVs (equal particle number as RT-BDEVs), along with increased gluconeogenic gene expression, suggesting that equal numbers of Cold-BDEVs could also facilitate hepatic gluconeogenesis (see Supplementary Figure S4A-S4C below; line 227-232, highlighted).

3) Similarly, equal amounts of Cold-BDEV protein (32 µg) treatment also resulted in decreased AKT phosphorylation, and consequently elevated gluconeogenic gene expression and hepatocyte glucose production after 48 h of incubation (see Figure 3A-D below; line 227-232, highlighted).

Collectively, our data showed that compared with hepatocytes treated with RT-BDEVs, either equal amounts of Cold-BDEV particles or protein treatment resulted in significantly elevated hepatic gluconeogenesis. Notably, when treated the BDEVs those secreted from the same amount of cultured BAT (closer to physiological changes), Cold-BDEVs led to a more significant enhancement of hepatic gluconeogenesis than RT-BDEVs.

6.-Treatment of mice with 80ug of BDEV has to be explained; please note that equal amounts of vesicles may have different amount of cargo proteins.

Answer: Thanks the reviewer for raising up this critical issue and we apologize for not clearly indicating why 80 μ g BDEVs were chosen to treat mice. In particular, we appreciate the reviewer for pointing out our neglectation that equal amounts of vesicles may have different amounts of cargo proteins, so even when administrating the same protein concentration of BDEVs, it is still a relatively rough quantification and normalization between groups. Therefore, in addition to treating mice with 80 μ g of BDEVs, which were calculated by the protein concentrations that secreted from cultured BAT as described below, we also conducted experiments using equal BDEVs particles between groups and BDEVs secreted from the same amount (100 mg) of BAT to mimic physiological conditions. The detailed experimental design and results are as follows:

1) As we mentioned in question 5, considering that the iBAT weight was similar between the control and cold-exposed mice (\sim 100 mg / mice), 100 mg of iBAT was isolated from control or cold-exposed mice for BDEV collection. After 24 h of incubation, \sim 80 μ g of RT-BDEVs ($\sim 4 \times 10^9$ particles) and \sim 160 μ g of Cold-BDEVs ($\sim 1 \times 10^{10}$ particles) were isolated from the culture medium of RT-BAT and Cold-BAT, respectively. Based on the results of the *in vitro* studies, we asked if equal protein amounts of Cold-BDEVs also promote hepatic gluconeogenesis in male mice. Therefore, 80 μ g RT-BDEVs or Cold-BDEVs were intravenously injected into mice each day for 3 days. Consistent with *in vitro* experiments, equal protein amounts of Cold-BDEV treatment resulted in markedly decreased levels of p-AKT, along with increased expression levels of gluconeogenic genes (see Figure 3H-I).

2) However, we did ignore that even equal amounts of vesicles may have different amounts of cargo proteins, so even administrating 80 μ g RT-BDEVs or Cold-BDEVs still does not indicate that equal amounts of BDEV vesicles were used. Therefore, equal amounts of RT-BDEVs or Cold-BDEVs (4×10^9 particles each day) were used to treat mice for 3 days in the revision. As indicated in Supplementary Figure S4D-S4F, decreased AKT phosphorylation and enhanced hepatic gluconeogenesis were also observed in mice treated with Cold-BDEVs. Moreover, as the cold-activated BAT secretes significantly more BDEVs than control BAT, to mimic the physiological conditions, the same treatment was conducted on mice using 80 μ g of RT-BDEVs ($\sim 4 \times 10^9$ particles) or 160 μ g of Cold-BDEVs ($\sim 1 \times 10^{10}$ particles), which were isolated from the same amount of the culture medium of RT-BAT and Cold-BAT. Not surprisingly, when treating BDEVs secreted from the same amount of cultured BAT, a more significantly enhancement of hepatic gluconeogenesis was observed in mice treated with Cold-BDEVs (see Supplementary Figure S4D-S4F below, line 242-246, highlighted). Therefore, by using different doses of BDEV treatment, we confirmed the function of cold-activated BDEVs in enhancing hepatic gluconeogenesis.

3) The absolute level of the important Cargo RNA in RT-BDEVs and in Cold-BDEVs were also measured. By using the standard curve to calculate the absolute level of miR-378a-3p, ~ 0.49 fmol miR-378a-3p was detected in each μg of RT-BDEV protein (~ 0.1 fmol miR-378a / 10^7 RT-BDEV particles), and ~ 1.75 fmol miR-378a-3p was detected in each μg of Cold-BDEV protein (~ 0.27 fmol miR-378a / 10^7 Cold-BDEV particles), respectively (see Supplementary Figure S6F below, line 290-293, highlighted). Therefore, exactly as the reviewer pointed, even equal amounts of BDEVs may have different amount of cargo proteins and RNAs. Here, the cargo RNA miR-378a-3p, was significantly increased in BDEVs after cold exposure.

Results:

7. Figure 1: please indicate tissue type for each analysis/graph indicating BAT or liver to make interpretation of the results easier for the reader.

Answer: Thanks the reviewer for pointing out this problem in Figure 1. We are sorry for the inconvenience the unclear labeling that has caused for the reader and have made corresponding labeling indicating tissue type for each graph in Figure 1 (see Figure 1B-D, 1G-H, 1I-J and 1P, and Supplementary Figure S1A-S1C).

8. Figure 3: 3A. Why did the authors incubate hepatocytes for 48 hours and 6 hours in Figure 2G?

Answer: Thanks the reviewer for raising this important issue. We completely understand that the different incubation times of BDEV with hepatocytes may cause confusion and apologize for not clearly explaining the reasons. As we described in Question 5, the main purpose in Figure 2G was to track whether BDEVs could be efficiently taken up by primary cultured hepatocytes. According to the time-course experiments, fluorescence signals could be detected in hepatocytes after 2 h of incubation, and the intracellular fluorescence intensity gradually increased and reached a plateau at 6 h. Therefore, 6 h of incubation was chosen for BDEV tracking in Figure 2A.

While for Figure 3A-D, we wonder whether BDEVs serve as modulators to regulate hepatocyte glucose metabolism. Usually, to ensure that the cells have had enough time to transcribe and translate genes, resulting in sufficient RNA and proteins for detection, functional detection is usually carried out after 24 or 48 h of treatment. Thus, in the preliminary study, the gene expression and protein levels of the hepatocytes were detected after 24 h and 48 h of coincubation with RT-BDEVs or Cold-BDEVs (32 μ g each, equal BDEV proteins were used for time point experiments here). After 24 h of incubation, p-AKT was decreased, concurrent with an elevation in gluconeogenic gene expression, while a more significant change ratio was observed after 48 h of coincubation (see Figure A-B below). Therefore, a 48h incubation time was chosen in Figure 3A-D and Supplementary Figure S4A-S4C to study the functional changes in hepatocytes.

9. How the authors checked if BAT-EVs reach other peripheral tissues or CNS? Does the cold-BDEV injection in Figure 3E may have a role on white adipose tissue in these animals?

Answer: Thanks the reviewer for bringing up this important issue regarding the possible target tissues of BDEVs. Currently, the targeting of EVs secreted from different tissues is still unclear, especially during physiological or pathological challenges. According to the reviewer's suggestion, by using Palm-GFP to track the Cold-BDEVs, we checked whether the BDEVs could reach other peripheral metabolic tissues and the brain during cold stress.

Of note, although the fluorescent GFP signals were highly expressed in the liver, the signals could also be detected in the kidney, heart, white adipose tissue (inguinal and epididymal adipose tissues) and skeletal muscle (see Supplementary Figure S3C below). However, we did not detect GFP signals in the arcuate nucleus of hypothalamus, hippocampus or cortex (see Supplementary Figure S3C below, line 207-211, highlighted).

As indicated by the *in vivo* EV tracking results, fluorescent signals were also detected in white adipose tissues, including epididymal and subcutaneous adipose tissues, suggesting that Cold-BDEVs may also regulate WAT functions. As Cold-activated BAT requires sharply increased uptake of both glucose and fatty acids as fuel supplies to generate heat, we asked whether Cold-BDEVs also regulate WAT lipolysis. As expected, compared with the mice treated with RT-BDEVs, Cold-BDEV treatment resulted in elevated mRNA levels of fatty acid oxidation genes *Cpt1a*, *Cd36* and *Acs1l*, suggesting that Cold-BDEVs may also play a role in facilitating WAT lipolysis (see Supplementary Figure S4G below, line 254-258, highlighted). These results indicate that cold-BDEVs may not only stimulate hepatic gluconeogenesis for glucose replenishment, but also play roles in regulating WAT lipolysis for fatty acid supply, which is quite reasonable for physiological regulation. We have discussed these interesting points in the revised discussion (line 443-445,

highlighted).

10. Have the authors consider to check this same mechanism in animals under high fat diet?

Answer: Thanks the reviewer for raising this important issue. In the current study, the experiments were conducted on the male mice (6-8 weeks old) fed a standard chow diet, they were supposed to represent the physiological regulatory mechanisms in normal mice. It is intriguing to know when the HFD-fed (pathologically changed) mice facing cold challenges, whether the same mechanism works or not under this pathological condition. Therefore, the 6-week-old male mice were housed at room temperature and fed a HFD (60% of calories from fat) for 10 weeks, followed by 72 h of cold exposure (see Figure A below). Notably, compared with control HFD-fed mice that housed at RT, cold-exposure resulted in significant elevations in gluconeogenic gene *G6pase* and *Pepck* (see Figure B below). Compared with HFD-fed control mice housed at RT, approximately 1.5-fold increase in miR-378a-3p was detected in the liver from cold-exposed HFD mice, without altering the transcription of miR-378a-3p (see Figure C-D below), indicating that BAT-derived miR-378a-3p may also be involved in this regulatory process. Therefore, BDEVs were isolated from the cultured BAT of RT-HFD mice or Cold-HFD mice. To mimic physiological conditions, total BDEVs collected from the cultured BAT (24 h incubation) were injected into each HFD mouse for 3 days. As expected, Cold-BDEVs increased the miR-378a-3p level and gluconeogenic gene expression (see Figure E-G below). While it is worth noting that although Cold-BDEV treatment also increased gluconeogenesis in the liver of HFD mice, the elevated ratio is relatively lower (~ 1.6-fold) than the changes in the liver from cold-exposed HFD-fed mice (~2 fold), suggesting that BDEV-derived miR-378a-3p partially contribute to the cold-induced hepatic gluconeogenesis in HFD-fed mice, and other regulatory mechanisms may also be involved in this process.

Reviewer #2 (Remarks to the Author):

Brown adipose tissue (BAT) takes up a lot of circulating glucose to fuel thermogenesis under cold exposure. However, little is known about the underlying mechanism. In this manuscript, the authors showed that cold stress stimulated BAT to produce more extracellular vesicles (BDEVs), selectively packaged more miR-378a-3p, a known metabolism-relevant miRNA encoded in the Pparg1b gene, into the EV, and delivered to the liver to promote gluconeogenesis. The authors confirmed miR-378a-3p enhances gluconeogenesis by targeting p110 α , a subunit of PI3K, which has been well-studied in a 2014 paper (ncomms6684). Further, the authors confirmed that miR-378 KO mice display reduced hepatic gluconeogenesis during cold exposure, while restoration of miR-378a-3p in iBAT induces the hepatic expression of gluconeogenic genes. These findings did provide some novel insight into metabolic interorgan coordination. However, there are still some major concerns that need to be addressed.

1. It is unclear if the other components besides miR-378a-3p also contribute to elevated hepatic gluconeogenesis. How much does miR-378a contribute to the hepatic gluconeogenesis effect of Cold-BDEV? The primary hepatocytes should be treated with Cold-BDEV and different concentrations of miR-378a-3p inhibitor to check how the hepatic gluconeogenesis will be affected.

Answer: Thanks the reviewer for giving us this excellent comments. We completely understand the reviewer's concern that other components besides miR-378a-3p may also contribute to the elevated hepatic gluconeogenesis, so it is important to directly demonstrate the role and contribution of miR-378a-3p in Cold-BDEVs in facilitating the hepatic gluconeogenesis. Therefore, based on the reviewer's constructive comments, primary hepatocytes were treated with Cold-BDEVs and different concentrations of miR-378a-3p inhibitor (antimiR-378a-3p) to check how the hepatic gluconeogenesis was affected. The conducted experiments and results are as follows:

The Cold-BDEVs were isolated from the cultured iBAT (100 mg) from cold-exposed mice (Line 157-158 in Methods, highlighted). 4×10^6 primary cultured hepatocytes were transfected with different doses of anti-miR-378a (0 nM, 10 nM, 25 nM, 50 nM, 100 nM, 200 nM) and co-incubation with 32 μ g Cold-BDEVs for 48 h. As an important target of miR-378a-3p in hepatocytes (Liu W et al, *Nat Commun.*2014 Dec 4;5:5684. PMID: 25471065), the protein level of p110 α was inversely correlated with the expression of the miR-378a-3p, thus, the p110 α protein level was used as a functional readout for different doses of anti-miR-378a. As expected, when doses of the anti-miR-378a gradually increased, the protein level of p110 α was also elevated in a dose-dependent manner, and reached the maximal level at 100 nM treatment and maintained thereafter (even 200 nM treatment could not cause further induction), suggesting that 100 nM antimiR treatment was sufficient to completely suppress the expression of miR-378a in hepatocytes co-incubated with Cold-BDEVs (see Figure 5G-5H below). In line with the gradually increased protein levels of p110 α , the expression of gluconeogenic genes was downregulated in a dose-dependent manner and decreased to the lowest level at 100 nM treatment (see Figure 5I below, line 331-339). These data suggested that miR-378a-3p in Cold-BDEVs plays an important role in regulating hepatic gluconeogenesis and affects the hepatic gluconeogenesis in a dose-dependent manner.

Figure 5. (G, H) Western blotting and densitometry analysis of p110α in hepatocytes. (n=6, 6). **(I)** Relative mRNA expression of gluconeogenic genes in hepatocytes (n=3 biological replicates). Statistical analysis was performed using one-way ANOVA followed by Bonferroni's multiple comparisons test. *P<0.05; **P<0.01; ***P<0.001.

In addition, to further determine the possible contribution of miR-378a-3p to Cold-BDEVs, by taking advantage of the high dose of anti-miR-miR-378a-3p (to completely blocked the miR-378-3p expression in treated hepatocytes), hepatocytes were treated with 100 nM anti-miR-378a-3p and co-incubated with/without Cold-BDEVs. In hepatocytes untreated with anti-miR-378a, Cold-BDEVs administration significantly reduced p110α protein level and markedly increased gluconeogenic gene expressions (~2-fold increase) (see Supplementary Figure S7A-S7B below, line 339-343, highlighted). While in those cells treated with anti-miR-378a (100 nM), comparable protein levels of p110α were detected in hepatocytes treated with or without Cold-BDEVs, suggesting that the function of miR-378a-3p derived from Cold-BDEVs were completely abolished in hepatocytes (see Supplementary Figure S7C below). Thus, by ruling out the affection caused by miR-378a-3p in cold-BDEVs, we checked whether other components in cold-BDEVs also contribute to hepatic gluconeogenesis. As shown in Supplementary Figure S7D, compared with control hepatocytes that were administrated high dose of anti-miR-378a-3p (100 nM), only slightly increased (~1.2-fold upregulation) expression of gluconeogenic genes was observed in cells treated with both anti-miRs and Cold-BDEVs, strongly indicating that other components may not be the major contributors in promoting gluconeogenesis (line 344-350, highlighted).

Supplemental Figure S7. (A) Western blotting and densitometry analysis of p110 α in hepatocytes. (n=6,6); **(B)** Relative mRNA expression of gluconeogenic genes in hepatocytes (n=3 biological replicates). **(C)** Western blotting and densitometry analysis of p110 α in hepatocytes. (n=6,6); **(D)** Relative mRNA expression of gluconeogenic genes in hepatocytes (n=3 biological replicates). Data presented as mean \pm s.e.m. Statistical analysis was performed using two-sided unpaired t-test. *P<0.05; **P < 0.01; ***P < 0.001.

Collectively, these data suggested that as a cargo RNA in cold-BDEVs, miR-378a-3p plays a very important role in promoting hepatic gluconeogenesis and functions in a dose-dependent manner.

2. *What will be the possible reason leading to the distinct responses of miR-378a-3p transcription in BAT and the liver to cold treatment?*

Answer: Thanks the reviewer for bringing up this important issue regarding the distinct responses of miR-378a-3p transcription in BAT and liver to cold exposure. As the transcriptional regulation of the same miRNA may vary among different tissues, we consulted the literature to find genes that have been reported to regulate miR-378a-3p in BAT or the liver, followed by experimental verifications.

1) As an enriched miRNA in BAT, miR-378a was reported to be positively regulated by *Ffar4* (a functional receptor for *n-3* polyunsaturated fatty acids) and promoted UCP1 expression (Pan D et al., *Nat Commun.* 2014 Aug 22; 5:4725.PMID:25145289; Kim J et al, *J Biol Chem.* 2016 Spe 23; 291(39): 20551-62. PMID:27489163). Specifically, the signaling axis *Ffar4*-miR-30b/378-Ucp1 was linked with the elevation of cAMP in brown adipocytes, which was similar to cold-exposed BAT (Kim J et al, *J Biol Chem.* 2016 Spe 23; 291(39): 20551-62. PMID:27489163). Therefore, the mRNA level of *Ffar4* was checked in the BAT of cold-exposed mice first. As shown in Supplementary Figure S6D below, the mRNA level of *Ffar4* was significantly increased in BAT when exposed to cold environment, consistent with the elevated transcription level of primary miR-378a in BAT (Figure 4I below), indicating that during cold exposure, elevated *Ffar4* may lead to increased miR-378a-3p transcription and expression in BAT. We also asked whether the same mechanism exists in livers from mice under cold stress. As shown in Supplementary Figure S6D below, decreased expression level of *Ffar4* was detected in the liver from cold exposed mice, indicating that upon cold exposure, the *Ffar4*-miR-378a-3p regulatory axis may only exist in the BAT but not the liver (line 281-285, highlighted).

2) Considering that the detected expression of *Ffar4* in the liver was much lower in the liver than in BAT, it may not represent the major transcription regulator of miR-378a in the liver. Therefore, another gene, *Lxr* (liver X receptor α), which has been reported to be upregulated in the nonalcoholic fatty liver and serves as a transcription activator of miR-378a (Zhang T et al, *Hepatology* 2019. Apr; 69 (4): 1488-1503. PMID: 30281809), was also checked here. As shown in Supplementary Figure S6E below, compared with the control group, the mRNA level of *Lxra* was decreased in the liver from cold-exposed mice, which was consistent with the decreased transcription level of miR-378a-3p (Figure 4I below), indicating that the transcription level of miR-378a-3p may also be regulated by *Lxra* in the liver during cold exposure, thereby the increased mature miR-378a-3p expression in the liver was unlikely from hepatic *de novo* synthesis (line 285-289, highlighted).

Collectively, these data suggested that in response to cold stimulation, the transcription of miR-

378a-3p in the BAT and liver may be differentially regulated by *Ffar4* and *Lxra*, respectively. However, other functional transcription activators or inhibitors of miR-378a may also participate in this regulation, which is worth further study in future work.

3. If the EV inhibitor GW4849 is systematically or BAT locally injected, how will the hepatic miR-378a-3p level change after the cold treatment or fasting?

Answer: Thanks the reviewer for pointing out this important issue. Considering that GW4869 inhibits EV release, systemic injection is expected to inhibit the EVs secreted from all kinds of organs/cells from the mice, while iBAT local administration is expected to mainly inhibit the EVs secreted from iBAT. Therefore, the different methods of GW4869 treatment may help us further understand how the miR-378a-3p be regulated in the liver under different physiological conditions (cold or fasting). Based on the reviewer's constructive comments, the cold exposed or fasted mice were subjected to either systematical (intraperitoneal) or BAT local injection of GW4869, the detailed experiments and results were as follows:

First, 100 µl of GW4869 (3 mg/kg) was locally injected into the BAT of cold-exposed mice by 5-8 point-injections each day for 3 days, followed by detection of miR-378a-3p level in the liver. After three days of injections, the mice were sacrificed 4 h after the last injection. Compared with cold exposed group, BAT locally treatment of GW4869 resulted in decreased level of miR-378a-3p in the liver upon cold exposure, without changing the transcription of miR-378a-3p (Pri-miR-378), suggesting that during cold exposure, the miR-378a-3p in the liver was affected by the Cold-BAT secreted EVs (see Figure 4K-4M below).

Figure 4. (K-M) The cold-exposed mice were injected *in situ* with either GW4869 or control vehicle. Mice were fasted for 4 h before sacrifice. Schematic diagram (**K**); qPCR analysis of the relative miR-378a-3p level (**L**) and primary miR-378a-3p level (**M**) in the liver. (n=8 each group, from 2 independent experiments). Data presented as mean \pm s.e.m. Statistical analysis was performed using two-sided unpaired t-test. *P<0.05; **P < 0.01; ***P < 0.001.

Next, 100 μ l of GW4869 (3 mg/kg) was intraperitoneally (systematically) injected into the mice cold-exposed mice each day for 3 days. Systematically treatment of GW4869 also resulted in decreased miR-378a-3p level in the liver (see Figure A-C below). Notably, compared with the locally administrated group, mice with systematically GW4869 treatment showed slightly lower level miR-378a-3p in the liver. This seems reasonable as i.p. GW4869 treatment was supposed to inhibit the EV secretion from all the tissues/cells, including all the brown adipose tissues, whereas interscapular BAT (iBAT) locally GW4869 administration mainly inhibited the EV secretion from iBAT. Although iBAT was the most abundant BAT in mice, small amount of BAT had also been detected in other regions, such as perirenal BAT, may also activated and contributed to the increased miR-378a-3p secretion. However, considering systematic GW4869 treatment is expected to inhibit the EV secretion from all kinds of organs/cells, it did seem to have a much broader influence on the mice metabolism, because during cold exposure, the food intake was reduced after i.p. GW4869 administration. Thus, in current study, local iBAT GW4869 treatment was chosen.

(A) Schematic diagram of GW4869 treatment. **(B)** qPCR analysis of the relative levels of miR-378a-3p in the liver. **(C)** qPCR analysis of the relative levels of primary miR-378a-3p in the liver. (n=8 each group, from 2 independent experiments) Data presented as mean \pm s.e.m. Statistical analysis was performed using two-sided unpaired t-test. *P<0.05; **P < 0.01; ***P < 0.001.

Furthermore, to explore the miR-378a-3p level change in mice subjected to overnight fasting for 16 h, 100 μ l of GW4869 (3 mg/kg) was also either systematically (intraperitoneally) or locally injected into the BAT 4 h before sacrifice. In line with previous reports, the hepatic miR-378a-3p level was elevated after overnight fasting (Liu W et al, *Nat Commun.*2014 Dec 4;5:5684. PMID: 25471065) (see Supplementary Figure S6G below), concurrent with increased transcription level of miR-378a-3p in the liver after fasting (see Supplementary Figure S6H below), indicating that the observed elevation of miR-378a-3p in the fasted liver was result from the hepatic *de novo* synthesis itself. It is worth noting that in cold-exposed mice, although miR-378a-3p expression was increased in the liver (see Figure 4H below), its transcription in the liver was decreased (see Figure 4I below), which was different from fasted mice.

Not surprisingly, neither systematically nor local treatment of GW4869 significantly altered the expression level of miR-378a-3p (see Supplementary Figure A-C and Figure S6I-S6K below), which is reasonable as the blockage of EV release may not affect the hepatic miR-378a-3p transcription and expression levels.

Collectively, these data strongly suggested that during cold exposure, the elevated miR-378a-3p in the liver mainly comes from the delivery of EVs from BAT, while during fasting, the elevated miR-378a-3p in the liver may result from the increase in its own transcription level in the liver (line 295-302, highlighted).

4. The AAV to specifically binding miR-378a-3p in BAT, such as AAV-FABP4-miR-378a-sponge, should be constructed, produced, and injected into the BAT and then check if the hepatic gluconeogenesis will be regulated or not.

Answer: We appreciate the reviewer for giving us this excellent comment. In the revision, we investigated whether specifically binding miR-378a-3p in BAT affected hepatic gluconeogenesis. According to the reviewer's suggestion, an adeno-associated virus expressing the miR-378a-3p-sponge driven by the FABP4 promoter (AAV9-FABP4-miR-378a-sponge, AAV^{FABP4}-sponge) was constructed, amplified and purified by HANBIO Biotechnology Co. Ltd (Shanghai, China). Briefly, the miR-378a-sponge sequence (CCTTCTGACTGGAGTCCAGTtatacCCTTCTGACTGGAGTC CAGTatcCCTTCTGACTGGAGTCCAGTtctcaCCTTCTGACTGGAGTCCAGT) was cloned

into the pHBAAV-FABP4-ZsGreen vector. After sequencing confirmation, the shuttle vector pHBAAV-FABP4-378a sponge-ZsGreen was packaged into capsids form AAV serotype 9 (AAV9-FABP4-miR-378a sponge-ZsGreen). Viral particles were purified and then titered by qRT-PCR analysis. For iBAT-specific knockdown of miR-378a-3p, 50 μ l of 1×10^9 Vg/ μ l of AAV9-FABP4-miR-378a-3p-ZsGreen (AAV^{FABP4}-sponge) was injected into the iBAT of mice. The corresponding control mice were injected with the same amount of NC vector (line 664-680 in Methods section, highlighted). As the schematic diagram indicates, 25 days post infection, mice were cold-exposed for 3 days prior to sacrifice (see Figure 8D). Compared with the control group (AAV^{FABP4}-NC), the miR-378a-3p was decreased in BDEVs, serum-EVs and the liver of mice infected with AAV^{FABP4}-sponge, while the transcription of miR-378a-3p in the liver was unaffected (see Figure 8E-8H below), indicating that binding of miR-378a-3p in BAT resulted in reduced EV-mediated miR-378a-3p secretion into the liver. Inhibition of miR-378a in iBAT did not alter the body weight, food intake or insulin level in cold-exposed mice (see Figure 8I-8K below). Knockdown of miR-378a in iBAT significantly increased the protein levels of p110 α and p-AKT in the liver, along with repressed hepatic gluconeogenic gene expression (see Figure 8L-8N below), thereby providing direct evidence that that inhibition of miR-378a in BAT could suppress cold-induced hepatic gluconeogenesis. Taken together, these data indicate that specifically binding miR-378a-3p in BAT led to decreased hepatic gluconeogenesis upon cold exposure (line 406-417, highlighted).

Figure 8. (D-N) iBAT-specific reduction of miR-378a-3p induced with adeno-associated virus constructed under the adipose tissue-specific promoter FABP4 (AAV^{FABP4}-sponge), a total of 50 μ l of 1×10^9 Vg/ μ l of each AAV was injected into the iBAT *in situ* of mice. Control mice were injected with AAV9- scrambled Control (NC). qPCR analysis of the relative miR-378a-3p expression level in (E) BDEVs (n=7,7); (F) serum EVs(n=6,6); and (G) livers (n=7,7). (H) Relative level of primary miR-378a-3p in the liver (n=6 each group, from 2 independent experiments). (I) body weight(n=7,7); (J) food intake(n=6,6); (K) serum insulin level (n=7 each group, from 2 independent experiments). (L) qPCR analysis of the mRNA levels of gluconeogenic genes in the liver (n=8 each group, from 2 independent experiments). (M) Western blotting for p-AKT, AKT, p110 α and α -Tubulin in the liver and (N) densitometry analysis of p-AKT, and p110 α . α -Tubulin was used as an internal control (n=6 each group, from 2 independent experiments). Mice were fasted for 4 h before test. Data presented as mean \pm s.e.m. All statistical analysis was performed using two-sided unpaired t-test. *P<0.05; **P<0.01; ***P<0.001.

5. Fig 4K, it should be critical to know the absolute level of miR-378a-3p (such as the counts per μ g of mass) included in the BDEV which was injected in the mice and the absolute level of miR-378a in the liver before and after the EV injection.

Answer: Thanks the reviewer for pointing out this critical issue. According to the reviewer's excellent suggestion, the absolute levels of miR-378a-3p in the BDEVs and the liver before and after the BDEV injection were calculated as follows:

In Figure 4J, equal protein amounts of RT-BDEVs or Cold-BDEVs (80 μ g) were intravenously injected into mice for three days, followed by miR-378a-3p detection in the liver 12 h after the last injection. By using the standard curve to calculate the absolute level of miR-378a-3p in each kind of BDEVs, \sim 0.49 fmol miR-378-3p was detected in each μ g of RT-BDEV protein (0.49 fmol miR-378a / μ g RT-BDEV), while \sim 1.75 fmol miR-378a-3p was detected in each μ g of Cold-BDEV protein (1.75 fmol miR-378a / μ g Cold-BDEV) (see Supplementary Figure S6F below, line 290-293, highlighted). Thus, for 80 μ g of RT-BDEVs and Cold-BDEVs those injected each day, the absolute miR-378a-3p included in 80 μ g RT-BDEVs was 39.2 fmol, while 80 μ g Cold-BDEVs included 140 fmol miR-378a-3p. In addition, the BDEVs were injected for 3 days, so the total treated amount of miR-378a-3p in RT-BDEVs was \sim 118 fmol, while for Cold-BDEVs, it was \sim 420 fmol.

Meanwhile, the absolute miR-378a-3p in the liver before and after EV injection was also calculated. As shown in Figure 4J below, \sim 79 fmol miR-378a-3p was detected per gram of liver mass before injection, and the total liver weight was \sim 1.15 g each mouse. Thus, the basal amount of miR-378a-3p before injection was \sim 79 fmol miR-378a / g liver (total \sim 90.9 fmol each liver). Three days after BDEVs injection, the miR-378a-3p level was determined 12 h after the last injection in the liver. As shown in Figure 4J below, \sim 102 fmol miR-378a /g liver (total \sim 117.3 each liver) was detected in the liver of the mice treated with RT-BDEVs, while for the Cold-BDEV treated group, \sim 182 fmol miR-378a /g liver (total \sim 209.3 per liver) was detected in the liver 12 h post the last injection (line 293-295, highlighted). Therefore, consistent with the previous relative ratio level, compared with the RT-BDEV group, a \sim 2 fold increase in miR-378a-3p was observed in the liver.

Collectively, based on the above calculation, we can see that the 3 days total amount of the given miR-378a-3p in the BDEVs, were generally higher than the detected elevated amount of miR-378a-3p in the liver. We thought it quite reasonable for the following reasons: 1) This reflected the final concentration of the miR-378a-3p elevation in the liver after 3 injections, which may be involved in a comprehensive kinetic change in both BDEVs and miR-378a-3p in the liver; 2) although most of the i.v. injected BDEVs enriched in the liver, some of the BDEVs still targeted other organs. Our appreciation for the reviewer’s excellent suggestion, according to calculation of the absolute miR-378a-3p concentrations in both BDEVs and the liver before and after injection, collectively, our data showed that compared with the RT-BDEV-treated group, the miR-378a-3p level was significantly increased after Cold-BDEV administration (see Supplementary Figure S6F and Figure 4J above, line 290-295, highlighted).

6. Line 252, “p110α, the fundamental subunit of phosphoinositide 3-kinase (PI3K) that controls glucose metabolism was predicted to be a direct target of miR-378a-3p.” However, in Ref 34 (), their title has clearly shown that “miR-378 targets p110α and controls glucose and lipid homeostasis”. p110α is a known target of miR-378 that doesn’t need to be predicted.

Answer: We appreciate the reviewer for pointing out this mistake and our sincere apology for the incorrect description. We have corrected the sentence into “p110α, the fundamental subunit of phosphoinositide 3-kinase (PI3K) that controls glucose metabolism, was demonstrated to be a direct target of miR-378a-3p in the previous study (Liu W et al, *Nat Commun.*2014 Dec 4;5:5684.PMID: 25471065)” (Line 326-328, highlighted).

Minor:

1. Fig 2I, it is unclear if other components besides EV will be transferred to the liver after the transduction of FABP4-Palm-EGFP lentivirus in BAT. QPCR should be performed to confirm the

GFP mRNA is detectable in the liver after the transduction.

Answer: Thanks the reviewer for pointing out this concern. We completely agree with the reviewer that GFP mRNA should be checked in the liver after virus transduction. In the revision, GFP mRNA was analyzed by QPCR in the BAT and liver after FABP4-Palm-EGFP lentivirus transduction. As shown in supplementary Figure S3B below, after lentivirus transduction, although significantly increased in mRNA level of GFP was detected in iBAT, GFP mRNA was still undetected in the liver (line 202-203, highlighted).

2. *Fig 2J-K, a BAT or liver-specific antibody should be used for co-immunostaining.*

Answer: We appreciate the reviewer for the excellent suggestion. UCP1 and CK18, which are expressed abundantly in brown adipocytes and hepatocytes, respectively, were commonly used as specific marker for BAT and liver (Colleluori et al., *Methods Mol Biol.* 2022; 2448:19-22. PMID: 35167088; Sato et al, *Blood.* 2005 Jul 15;106(2):756-63. PMID: 15817682). Therefore, UCP1 and CK18 were coimmunostained with the BAT and liver, respectively. As indicated in Figure 2J-K below, coimmunostaining of UCP1 (BAT marker, red) and GFP (green) confirmed that cold exposure resulted in a denser appearance and higher UCP1 expression in iBAT, with approximately 70% BAT infection in both groups. Notably, the appearance of GFP signals in the liver suggested that cold exposure resulted in significantly increased BDEV uptake by hepatocytes (CK18-positive, red) compared to RT group (see Figure 2J-K below, line 196-202, highlighted).

3. Fig 2H, the palmitoylated residue should be cysteine but no other amino acids, as in as in <https://www.nature.com/articles/ncomms8029> Fig1a.

Answer: We are very grateful to the reviewers for pointing out the mistake. As the reviewer mentioned, the palmitoylated residue was cysteine, and we apologize for the mistaken label in the schematic in Figure 2H. We have corrected this mistake in the revision (see Figure 2H below, line186, highlighted).

4. Fig 4H-I, it is unclear if the level or activity of the miRNA biogenesis proteins such as Dicer will be elevated by cold in the mouse liver, leading to the increase of mature miRNAs level.

Answer: We are very grateful to the reviewers for raising up this important issue. Based on the reviewer's constructive comments, the mRNA and protein levels of Dicer were first checked. As shown in Supplementary Figure S6L-S6M below, compared with the expression level in mice housed at RT, cold exposure did not alter the transcription and translation of Dicer in the liver. We completely agree with the reviewer that the elevation of the level or activity of other genes related to miRNA biogenesis may result in an increased mature miRNA expression level. Therefore, the expression levels of several other miRNAs, including miR-192, miR-21, and miR-223, were also checked in the cold-exposed liver. As shown in supplementary Figure S6N below, the expression levels of miR-192 and miR-223 were unchanged in the cold-exposed liver, whereas the expression of miR-21 was decreased in the liver upon cold exposure. Taken together, these uneven changes in miRNA expression levels in the liver strongly suggested that the increased miR-378a-3p was unlikely due to the change in the miRNA processing proteins in the liver (line 302-307, highlighted).

5. The reference for GSE41306 should be cited.

Answer: Thanks the reviewer very much for pointing out the missing reference citation in the manuscript. We apologize for the error and have added the cited reference in the revision (Trajkovski et al, *Nat Cell Biol.*2012 Dec; 14(12):1330-5. PMID: 23143398. Line 269, 741, 793, 1002, highlighted).

6. Line 548, AAV is not adenovirus. How these AAV vectors were constructed should be demonstrated.

Answer: We appreciate the reviewer for pointing out this mistake. Our sincere apology for mistakenly describing the Adeno-associated virus (AAV) as adenovirus in the subtitle in the Method section. We have corrected it to “Adeno-associated virus recombination and administration” (Line 664 in Methods, highlighted). All adeno-associated virus (AAVs), including AAV9-FABP4-miR-378a-ZsGreen (AAV^{FABP4}-miR-378a), AAV9-FABP4-miR-378a sponge-ZsGreen (AAV^{FABP4}-sponge), and AAV8-TBG-miR-378a sponge-ZsGreen (AAV^{TBG}-sponge) were constructed, amplified, and purified by HANBIO Biotechnology Co. Ltd (Shanghai, China). Briefly, miR-378a-3p (AGGGCTCCTGACTCCAGGTCCTGTGTGTTACCTCGAAATAGCACTGGACTTGGAGT CAGAAGGCT) was cloned into the pHBAAV-FABP4-ZsGreen vector, followed by sequencing confirmation, then the shuttle vector was packaged into capsids from AAV serotype 9 (AAV9-FABP4-miR-378a-ZsGreen, used in BAT); miR-378a-sponge (CCTTCTGACTGGAGTCCAGT tatacCCTTCTGACTGGAGTCCAGT tacatcCCTTCTGACTGGAGTCCAGT tcttcaCCTTCTGACT GGAGTCCAGT) was cloned into the pHBAAV-FABP4-ZsGreen and pHBAAV-TBG-MCS-P2A-ZsGreen vectors, respectively. After sequencing confirmation, the shuttle vector pHBAAV-FABP4-378a sponge-ZsGreen was packaged into capsids form AAV serotype 9 (AAV9-FABP4-miR-378a sponge-ZsGreen, used in BAT), while the shuttle vector pHBAAV-TBG-MCS-P2A-ZsGreen was packaged into capsids from serotype 8 (AAV8-TBG-miR-378a sponge-ZsGreen, used in liver).

Viral particles were purified and then titered by qRT-PCR analysis (Line 665-680 in Methods, highlighted).

REVIEWERS' COMMENTS

Reviewer #1 (Remarks to the Author):

The authors have fully replied point by point this reviewer comments. Moreover, they performed adequate experimental work, that was added to the manuscript, supporting the results needed to response my questions and suggestions. I believe that this new version of the manuscript has improved appropriately by adding relevant data to the field.

I just have a minor comment regarding the response to comment 2-Figure 2B: Authors show BDEVs characterization as requested. However, authors should explain and discuss why they don't detect UCP1 in the isolated EVs since it has been described elsewhere that this protein is characteristic of BAT adipose tissue shed EVs (Camino et al., IJMS 2022), and it has been shown that brown adipocytes eject mitochondria in EVs (Starling S, Nature Reviews Endocrinology 2022).

Reviewer #2 (Remarks to the Author):

Thank you for thoroughly addressing all of my previous concerns in the initial review. Your team has done an excellent job in studying the unique BAT-liver crosstalk mechanism, specifically the EV-mediated miR-378-3p transfer under cold treatment. I have no further comments.

Reviewer #1 (Remarks to the Author):

The authors have fully replied point by point this reviewer comments. Moreover, they performed adequate experimental work, that was added to the manuscript, supporting the results needed to response my questions and suggestions. I believe that this new version of the manuscript has improved appropriately by adding relevant data to the field.

I just have a minor comment regarding the response to comment 2-Figure 2B: Authors show BDEVs characterization as requested. However, authors should explain and discuss why they don't detect UCP1 in the isolated EVs since it has been described elsewhere that this protein is characteristic of BAT adipose tissue shed EVs (Camino et al., *IJMS* 2022), and it has been shown that brown adipocytes eject mitochondria in EVs (Starling S, *Nature Reviews Endocrinology* 2022).

Answer: Thank you for pointing out this critical issue. As mentioned by the reviewer, the previous study by Camino et al. reported a significant up-regulation of UCP1 protein level in obese BAT-derived EVs (Camino et al., *Int J Mol Sci.* 2022 Sep 16;23(18).10826. PMID: 36142750). Additionally, Rosina et al. demonstrated that adrenergically stressed brown adipocytes ejected damaged components of mitochondria via EVs (Rosina et al., *Cell Metab.* 2022 Apr 5; 34(4):533-548. PMID: 35305295). However, in our findings presented in supplementary Figure S2e, the UCP1 protein that enriched in the BAT, was detected in the RT-BAT-derived medium/large EVs (M/L-EVs) with a size > 200 nm. While we did not detect UCP1 from the same amount of smaller sized RT-BDEVs (~100 nm, Supplementary Figure S2e). Therefore, we fully acknowledge the need to explain and discuss these seemingly contradictory results. We carefully compared the extraction methods and characterization used in the aforementioned BAT-EV studies. To further investigate the presence of UCP1 in BAT-derived EVs isolated from mice, we conducted additional western blot analyses, examining different protein concentrations of BDEVs (with size ~100 nm). According to the comparison and data analysis, we believe that the observed differences could be attributed to variations in the animal models employed, the methods used for EV extraction, the methodology used for selection and comparison of EV proteins, metabolic conditions, the specific EV subpopulations tested, and especially the different amounts of EVs used for immunoblot detection in various studies. The detailed comparisons and analyses are described below:

1. Various animal models were used in the above studies: *Camino et al.* isolated BAT-EVs from lean and HFD-induced obese male Sprague-Dawley rats, while *Rosina et al.* extracted BAT-EVs from male C57BL/6 mice exposed to cold (4 °C for 20 h) or acclimatized to thermoneutrality (30°C, TN). In our study, the BDEVs were isolated from male C57BL/6 mice exposed to cold (4 °C for 72 h) or housed at room temperature (25 °C, RT), which were similar with the animal models those used in *Rosina et al's* study.

2. Different methods were used to extract BAT-EVs in the above studies. In *Camino's* study, 1g BAT from either lean or obese rats was extracted and cultured in DMEM for 48 h to isolate EVs. The collected BAT secretomes underwent initial filtration using a 0.22 µm filter to eliminate contaminating cell debris. Subsequently, centrifugation was carried out at 10,000 × g at 4 °C for 20 min and 90 min to pellet the EVs. The resulting EV pellet was then resuspended in ice-cold PBS and subjected to a second round of ultracentrifugation (10,000 × g at 4 °C for 90 min). The resulting

EV pellet was used for subsequent analysis.

In *Rosina's* study, approximately 125 mg of BAT from either TN or cold-exposed mice was cultured in DMEM for 24 h to extract EVs. The culture medium was collected and centrifuged at $600 \times g$ for 10 min at 4 °C to eliminate dead cells. Subsequently, the supernatant was underwent centrifugation at $17,000 \times g$ at 4 °C to isolate large EV fractions (Large-EVs), or it was ultracentrifuged at $100,000 \times g$ for 16 h at 4 °C to obtain the total EV fraction.

In our study, 100 mg iBAT from either cold-exposed or control mice was excised and incubated in DMEM for 24 h. The culture supernatant was then centrifuged at $300 \times g$ for 10 min, $3,000 \times g$ for 20 min, and $10,000 \times g$ for 30 min to remove tissues, cells, debris and medium/large EVs. The resulting supernatant was filtered through a 0.22 μm filter and ultracentrifuged at $110,000 \times g$ for 70 min at 4 °C to pellet EVs. The EV collections were washed once, resuspended in PBS and subjected to a second round of centrifugation at $110,000 \times g$ for 16 hours at 4 °C. Finally, the EV pellet was resuspended in DMEM or PBS.

Based on the above description, we can see that the EV isolation methods differed among the three groups. Generally, although not identical, our extraction method resembled that of *Rosina et al.'s* study. Consequently, the variations in the steps and the speeds and duration of centrifugation might influence the subpopulation of the collected EVs.

3. Variations in the characterization of EV size were observed across different studies. In *Camino's* study, EVs extracted from the rat BAT samples using nanoparticle tracking analysis (NTA) ranged in size from 75 to 112 nm. In *Rosina's* study, EV diameter measurements obtained through dynamic light scattering (DLS) revealed two main populations at approximately 350 nm and 50 nm, which likely represented microvesicles (MVs or medium/Large-EVs) and exosomes, respectively. In our study, NTA analysis revealed that the extracted BDEVs showed an enrichment of small EVs (<200 nm), with peak at around 100 nm. We also found that the isolated EVs ranged from 200 to 800 nm, peaking at approximately 230 nm, falling within the size range of Large/Medium-EV size (>200 nm) as defined by the International Society of Extracellular Vesicles (Théry et al., *J Extracel Vesicle*. 2018). Overall, consistent with the earlier observation on the use of varied BAT-EV extraction methods in the aforementioned studies, the discrepancies in the isolated BAT-EV size may indicate the collection of distinct subpopulations of BAT-EVs, potentially leading to diverse cargo protein/RNA enrichment.

4. Differences in the analytical methodology used for the selection and comparison of BAT-EV proteins as described below:

In *Camino's* study, the researchers conducted a proteomic analysis of BAT-EVs derived from 1g of cultured rat BAT tissue obtained from either obese or lean rats. Mass spectrometry data dependent acquisition (DDA) was employed for the proteomic analysis of BAT-EVs isolated from obese and lean rats. By comparing the proteins identified through DDA with the BATLAS database, which is based on the transcriptome signature of murine brown, brite and white adipocytes (*Perdikari et al*, cell Rep. 2018 Oct 16; 25(3):784-797. PMID: 30332656), UCP1 was identified as one of the proteins present in obese BAT-EVs and also exhibited significant elevation compared to EVs isolated from lean rat BAT. To confirm this finding, immunodetection was utilized to analyze the EV proteins secreted from 1g of BAT from obese and lean rats. Consistently, a considerably higher level of UCP1 protein was found in EVs secreted from obese BAT compared to those from

lean BAT.

In *Rosina's* study, the proteome of BAT-EVs was profiled using quantitative proteomics. These EVs were obtained from BAT samples of mice exposed to cold or acclimatized at thermoneutrality, with each sample containing 100 μg of BAT-EVs. By comparing the proteome of BAT-EVs with mitochondrial proteins (GO:0005739) and MitoCarta 3.0, enriched terms associated with mitochondrial components and metabolism were identified in cold BAT-EVs. Furthermore, by filtrating EVs using a 0.22 μm filter, they demonstrated the presence of extracellular mitochondrial components in the largest fraction of EVs (>200 nm), which were not present in exosomes (~50 nm). Importantly, their results revealed that UCP1, the most important thermogenic protein in BAT, was not up-regulated in the top 100 proteins identified in BAT-EVs, which was confirmed by western blot analysis. This was in contrast to the high expression of UCP1 in cold-exposed BAT mitochondria (3 μg BAT-mitochondria protein), where it was not detected in cold-exposed BAT-EVs (20 μg BAT-EV protein).

In our study, the presence of UCP1 protein, predominantly found in BAT, was examined in RT-BAT using western blot analysis (see Supplementary Figure S2e). The analysis revealed that UCP1 protein was detected in the medium/large EVs (15 μg , size > 200 nm) obtained from RT-BAT but not in the same quantify of small EVs (15 μg , ~ 100 nm), referred to as RT-BDEVs. This observation indicates that mitochondrial components are more likely to be released by larger-EVs in comparison to smaller EVs, which partially supports the results obtained in *Rosina's* study.

In addition to the aforementioned differences, we observed variations in the quantities of EVs used across these studies. Moreover, the metabolic conditions, such as obesity (pathological) or cold exposure (physiological), may contribute to disparate findings in terms of UCP1 expression in EVs. To investigate the presence of UCP1 in small EVs (referred to as BDEVs) isolated from mice housed at room temperature (RT), we conducted additional western blot analyses, examining different protein concentrations of RT-BDEVs with a size below 200 nm (peaked at ~100 nm). As shown in Figure A, a 5 μg RT-BAT protein sample served as positive control. Similar to previous observations (Supplementary Figure S2e), UCP1 was observed in the 5 μg Large-EV sample, while UCP1 detection in the same quantify of RT-BDEV (5 μg) samples was limited. However, notably, as the loaded RT-BDEV protein amount increased to 50 μg and 100 μg , UCP1 became detectable and the levels elevated accordingly. Densitometry calculations indicated that the UCP1 protein level in RT-BDEVs was approximately 1/100th compared to an equivalent protein amount from RT-BAT. Thus, this differences in UCP1 level in RT-BAT and in RT-BDEVs may surpass the detection sensitivity of immunoblot, necessitating a higher protein amount of RT-BDEVs for effective UCP1 assessment.

Figure A. Western blotting detection of nuclear protein LAMIN A/C and mitochondrial inner membrane protein UCP1 in the RT-BAT, RT medium/large EVs (M/L EVs) and RT-BDEVs. (n=6 per group of mice, from 2 independent experiment).

Interestingly, we found that the results presented above support findings from both studies. Firstly, in *Camino's* study, UCP1 protein was detected in BAT-EVs from lean rat, but with much higher levels in obese BAT-EVs. This suggests that a larger quantity of BAT-EVs might be required for efficient UCP1 detection in immunoblots of lean animals housed at RT. Secondly, as suggested by *Rosina's* study, we also found that the mitochondria components were more enriched in the larger fractions of RT-BAT-EVs in mice housed at RT (>200 nm).

Taken together, research interest on the metabolic regulatory role in BAT has significantly increased since the characterization of active BAT. Several studies have focused on BAT-derived EVs. However, a standardized methodological approach for BAT-derived EVs has yet to be fully established. In the present study, we found that UCP1 protein was not enriched in the small-sized RT- BDEVs (~100 nm) compared to medium/large EV fractions (>200 nm). We speculated that the observed differences among studies may be due to variations in animal models, metabolic conditions, EV extraction methods, EV protein selection and comparison methods, testing of specific EV subpopulations, and the amount of EVs used for immunoblot detection in different studies, which requires further study. We have included the description in the results (line 166-168) and discussed the issue in the revised discussion section (line 458-468 in Discussion section, highlighted).

Reviewer #2 (Remarks to the Author):

Thank you for thoroughly addressing all of my previous concerns in the initial review. Your team has done an excellent job in studying the unique BAT-liver crosstalk mechanism, specifically the EV-mediated miR-378-3p transfer under cold treatment. I have no further comments.

Answer: Thank you very much for taking the time to carefully review the manuscript and offer excellent comments that helps us to improve the manuscript.